# Aurora kinase A localises to mitochondria to control organelle dynamics and energy production

Giulia Bertolin[1,2]*, Anne-Laure Bulteau[3,4,5], Marie-Clotilde Alves-Guerra[6,7,8], Agnes Burel[9], Marie-Thérèse Lavault[9], Olivia Gavard[1,2,10,11], Stephanie Le Bras[1,2], Jean-Philippe Gagné[11], Guy G Poirier[11], Roland Le Borgne[1,2,10], Claude Prigent[1,2,10]*, Marc Tramier[1,2,9]*

[1]CNRS, UMR 6290, Rennes, France; [2]Université de Rennes 1, UBL, Genetics and Development Institute of Rennes (IGDR), Rennes, France; [3]ENS de Lyon, Lyon, France; [4]CNRS UMR 5242, Lyon, France; [5]INRA USC 1370, Lyon, France; [6]Inserm, U1016, Institut Cochin, Paris, France; [7]CNRS, UMR 8104, Paris, France; [8]Université Paris Descartes, Sorbonne Paris Cité, Paris, France; [9]Microscopy Rennes Imaging Centre, SFR Biosit, UMS CNRS 3480- US INSERM 018, Université de Rennes , Rennes, France; [10]Equipes labélisées Ligue Contre Le Cancer, Rennes, France; [11]Centre de recherche du CHU de Québec, Faculté de Médecine, Université Laval, Québec, Canada

*For correspondence:
giulia.bertolin@univ-rennes1.fr
(GB);
claude.prigent@univ-rennes1.fr
(CP);
marc.tramier@univ-rennes1.fr (MT)

**Competing interests:** The authors declare that no competing interests exist.

**Abstract** Many epithelial cancers show cell cycle dysfunction tightly correlated with the overexpression of the serine/threonine kinase Aurora A (AURKA). Its role in mitotic progression has been extensively characterised, and evidence for new AURKA functions emerges. Here, we reveal that AURKA is located and imported in mitochondria in several human cancer cell lines. Mitochondrial AURKA impacts on two organelle functions: mitochondrial dynamics and energy production. When AURKA is expressed at endogenous levels during interphase, it induces mitochondrial fragmentation independently from RALA. Conversely, AURKA enhances mitochondrial fusion and ATP production when it is over-expressed. We demonstrate that AURKA directly regulates mitochondrial functions and that AURKA over-expression promotes metabolic reprogramming by increasing mitochondrial interconnectivity. Our work paves the way to anti-cancer therapeutics based on the simultaneous targeting of mitochondrial functions and AURKA inhibition.
DOI: https://doi.org/10.7554/eLife.38111.001

## Introduction

The mitotic kinase AURKA controls centrosomal maturation, the timing of mitotic entry, bipolar and central spindle assembly, and cytokinesis (*Nikonova et al., 2013*). AURKA is over-expressed in epithelial cancers, and it is believed to act as an oncogene through the induction of genomic instability. AURKA alters the number of centrosomes, the properties of the mitotic spindles, it induces aneuploidy and a defective cell division (*Bischoff et al., 1998*; *Zhang et al., 2004*; *Nikonova et al., 2013*). To perform these functions, AURKA interacts with multiple proteins at centrosomes and on the mitotic spindle, in a kinase-dependent or independent manner (*Nikonova et al., 2013*). In mice, the over-expression of AURKA suffices to induce the appearance of mammary tumours similar to human breast cancers (*Wang et al., 2006*; *Treekitkarnmongkol et al., 2016*). Thus, AURKA has been of high interest for pharmaceutical companies as a drug target, despite the currently available

**eLife digest** Structures called mitochondria power cells by turning oxygen and sugar into chemical energy. Each cell can have thousands of mitochondria, which work together to supply changing energy demands. They can fuse together or break apart, forming networks that change size and produce different amounts of energy. Getting the balance right is crucial; if energy levels are too low, the cell will not be able to grow and divide. If energy levels are too high, the cell can grow at a faster rate, which can contribute to the cell becoming cancerous.

Although we know that mitochondria provide energy, it is not clear how they communicate to fine-tune the supply. Some clues come from cancer cells that seem dependent on their mitochondria for survival. In these cells, levels of a protein called AURKA are higher than normal. AURKA helps cells to divide, and it interacts with many different proteins. This complexity makes it difficult to work out exactly what AURKA does, but it is possible that it plays a role in energy supply.

Bertolin et al. have now investigated whether mitochondria use AURKA to communicate inside human breast cancer cells. Tagging AURKA proteins with a fluorescent marker revealed that it accumulates inside mitochondria. Once it gets there, AURKA changes the shape of the mitochondria, which has dramatic effects on their capacity to produce energy. At normal levels, AURKA causes the mitochondria to fragment, breaking apart into smaller pieces. This maintains their energy output at a normal level. If AURKA levels are too high, the mitochondria fuse together and produce more energy. This means AURKA could help to fuel fast-growing cancer cells.

Current drugs that aim to treat cancer by blocking the activity of AURKA show poor results. This is partly due to the fact that the protein has so many different roles in the cell. Finding that AURKA affects mitochondria is the first step in understanding one of its unknown roles. It also suggests the possibility of developing new drugs to change how mitochondria make energy in cancer cells that contain high levels of AURKA.

DOI: https://doi.org/10.7554/eLife.38111.002

drugs show only modest beneficial effects in patients. Although still at a preclinical level, the only promising strategy appears to be the combination of AURKA inhibitors with agents simultaneously targeting multiple cancer-relevant AURKA partners and functions (*Nikonova et al., 2013*).

Epithelial cancers were found to be dependent on mitochondrial ATP, produced through oxidative phosphorylation, as a source of energy (*Whitaker-Menezes et al., 2011*). Particularly, epithelial cancer cells were found to carry a specific molecular signature constituted by 38 genes regulating the functionality of the mitochondrial respiratory chain, the synthesis of mitochondrial ribosomes and the import of proteins into mitochondria. It is known that mitochondria adapt their ATP production rate by modulating their morphology from fragmented organelles to an interconnected network (*Mishra and Chan, 2016*). In particular, mitochondrial fusion was shown to increase ATP production in several paradigms (*Mitra et al., 2009*; *Tondera et al., 2009*), granting protection against apoptosis (*Lee et al., 2004*) and increasing cell proliferation (*Mitra, 2013*). However, how these effects are established by increasing mitochondrial connectivity remains to be mechanistically elucidated. In mammals, fusion is mediated by the Outer Mitochondrial Membrane GTPases MFN1 and 2 and the Inner Mitochondrial Membrane OPA1, which undergoes a proteolytic cleavage from a long (L-OPA1) to a short (S-OPA1) isoforms. The balance between L- and S-OPA1 is controlled by multiple mitochondrial proteases to ensure mitochondrial fusion both in normal and in stress conditions (*Mishra and Chan, 2016*). Conversely, mitochondrial fission is regulated by the cytosolic GTPase DNM1L through its interaction with several mitochondrial receptors (MFF, MIEF1 and 2, FIS1), but also through the regulation of its post-translational modifications (phosphorylation, SUMOylation, acetylation and S-nitrosilation) (*Mishra and Chan, 2016*). Mitochondrial fission is also used to separate dysfunctional mitochondria from the healthy network prior to their degradation by mitophagy, a selective type of autophagy (*Youle and Narendra, 2011*).

Given that AURKA is a hallmark of epithelial cancers, understanding the complexity of the AURKA interactome is mandatory to optimise therapeutic strategies in patients with epithelial cancers derived from AURKA over-expression. Despite often considered a mitotic protein, recent evidence showed that AURKA is active at interphase as well (*Mori et al., 2009*; *Bertolin et al., 2016*),

although its roles beyond mitosis are still largely unexplored. We here show that interphasic AURKA localises to mitochondria, where it is imported and processed. While exploring the potential roles of mitochondrial AURKA, we observed that it increases mitochondrial fusion through a direct interaction with proteins regulating mitochondrial dynamics. The modulation of mitochondrial fusion and fission mechanisms when AURKA is over-expressed increases ATP production *via* the mitochondrial respiratory chain.

## Results

### AURKA localises in the mitochondrial matrix *via* an N-terminal MTS and it undergoes a double proteolytic cleavage

While exploring the localisation of AURKA at interphase, we observed that AURKA co-localises with the mitochondrial processing peptidase PMPCB in human MCF7 cell lines (*Figure 1A*). The fluorescence signal of AURKA observed at mitochondria is specific, as it disappeared after AURKA knockdown by siRNA-mediated gene silencing (*Figure 1A* compare the two left panels and histograms). AURKA depletion also leads to profound changes in the organisation of the mitochondrial network, strongly suggesting a functional role of AURKA at mitochondria (*Figure 1A* compare the two middle panels). In addition, AURKA localises to mitochondria regardless of the cell cycle phase and of its relative abundance (*Figure 1—figure supplement 1A*).

We then explored how AURKA is imported into mitochondria. The vast majority of mitochondrial proteins undergo one or sequential proteolytic cleavages when imported into these organelles (*Chacinska et al., 2009*). First, the mitochondrial matrix peptidase PMPCB cuts the Mitochondrial Targeting Sequence (MTS) off the mitochondrial precursor protein (*Chacinska et al., 2009*). Then, a second or multiple mitochondrial proteases can further cleave the pre-protein and allow it to reach the mitochondrial sub-compartment of destination. We therefore searched for mitochondrial AURKA isoforms representing one or more cleavage products. In immunoblots of total cell and mitochondrial lysates from HEK293 cells, AURKA could be detected as three isoforms: a predominant full-length isoform of ~46 kDa ($AURKA_{46}$), an intermediate isoform of ~43 kDa ($AURKA_{43}$) compatible with a first proteolytic cleavage by PMPCB (*Chacinska et al., 2009*), and a short isoform of ~38 kDa ($AURKA_{38}$) that presumably represents the mature mitochondrial isoform and which is detectable with a monoclonal (*Figure 1B*) and a polyclonal anti-AURKA antibody (*Figure 1—figure supplement 1B*). These results also corroborate the mitochondrial localisation of AURKA in a second cell line. To confirm that $AURKA_{43}$ and $AURKA_{38}$ correspond to two AURKA isoforms processed inside mitochondria, we depleted 80% of the mitochondrial protease PMPCB. This almost completely blocked the intra-mitochondrial cleavage of AURKA (*Figure 1C*), strongly suggesting that $AURKA_{43}$ and $AURKA_{38}$ are mitochondrial cleavage products issued by PMPCB-related import pathways in the matrix. To identify the cleavage sites from where $AURKA_{43}$ and $AURKA_{38}$ originate we used nanoLC-ESI MS/MS to search for semi-tryptic peptides corresponding to the N-terminally processed isoforms $AURKA_{43}$ and $AURKA_{38}$. Compared to canonical peptides generated by the proteolytic action of trypsin, semi-tryptic peptides are cut non-canonically and they can therefore be generated by the action of mitochondrial proteases (*Vögtle et al., 2009*). We retrieved two semi-tryptic peptides starting at residues 33 and between residues 80–82; the size of these peptides was consistent with the molecular weights of $AURKA_{43}$ and $AURKA_{38}$, respectively (*Figure 1—figure supplement 1C* and *Supplementary file 1*).

We then investigated the exact sub-mitochondrial localisation of AURKA in HEK293 cells using transmission electron microscopy (TEM). We first detected ectopic AURKA fused to GFP in the mitochondrial matrix and in contact with mitochondrial *cristae* (*Figure 1D*). This localisation was confirmed by employing different antibody combinations (*Figure 1—figure supplement 1D and E*) and by illustrating that AURKA localises to the mitochondrial matrix similarly to the mitochondrial matrix protein SOD2 (*Figure 1—figure supplement 1F*). To determine whether AURKA is a soluble protein or it is strongly attached to mitochondrial membranes, protein extraction with sodium carbonate was carried out on mitochondrial fractions of HEK293 cells (*Figure 1E*). AURKA was retrieved exclusively in the soluble fraction, reinforcing the conclusion that AURKA is mainly localised in the matrix as observed in TEM analyses. To further investigate the import of AURKA, we digested mitochondrial fractions of HEK293 cells with trypsin to eliminate mitochondrial protein precursors and

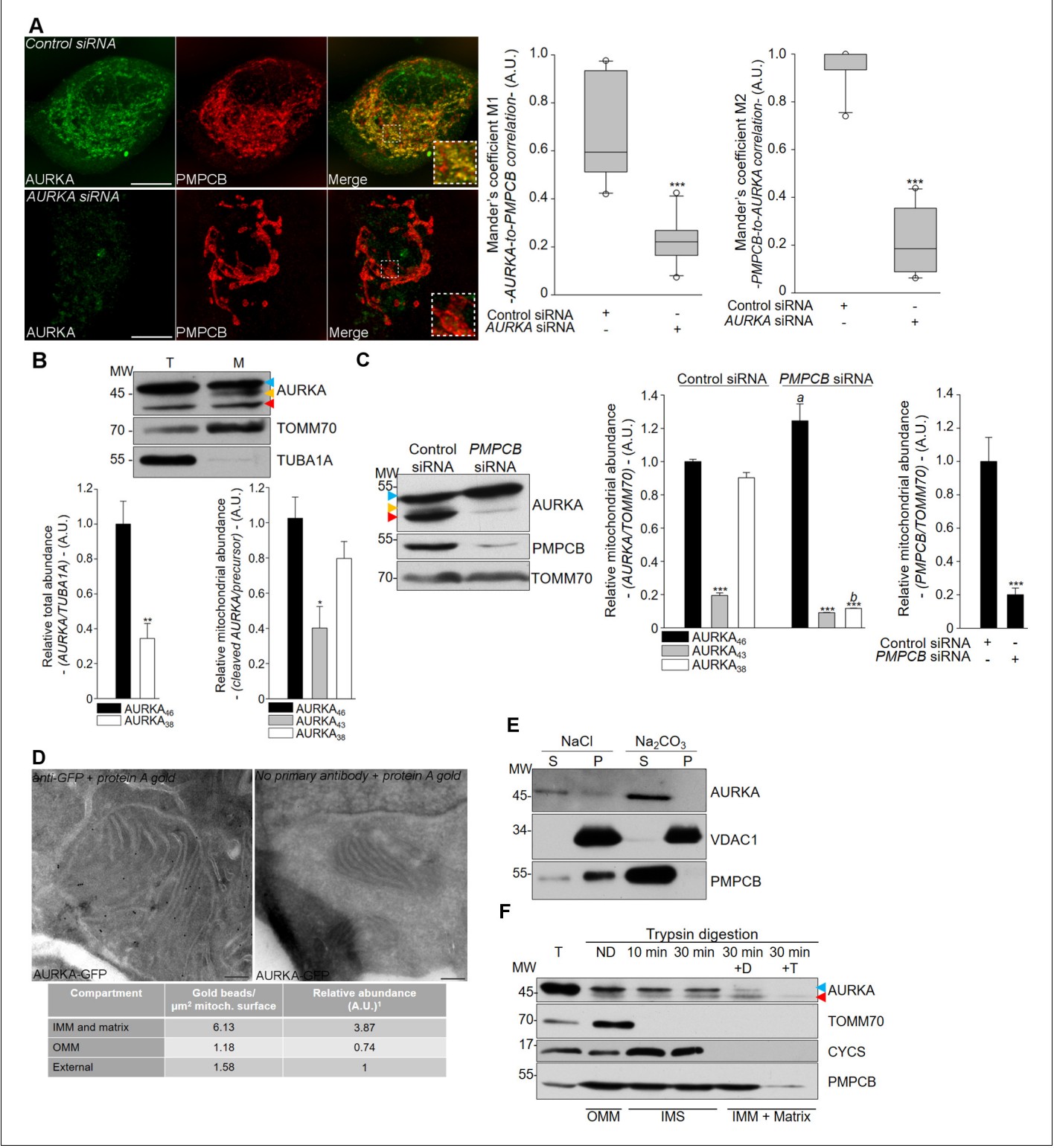

**Figure 1.** AURKA localises to mitochondria and it is imported into the mitochondrial matrix. (**A**) (Left) Immunofluorescence micrographs of MCF7 cells transfected with control (top panels) or AURKA-specific siRNA (bottom panels); cells were stained for endogenous AURKA (left panels) and with PMPCB (middle panels) for mitochondria. Inset: higher magnification of the dotted area. Scale bar: 10 μm. (Right) Mander's M1 and M2 co-localisation coefficients (***Bolte and Cordelières, 2006***) between AURKA and PMPCB on confocal pictures as in (**A**). n = 10 cells per condition; one representative experiment (of three) is shown. Whiskers extend from the 5th to the 95th percentiles. Outliers are indicated by white dots. (**B**) (Top) Lysates from total (**T**) and mitochondrial (**M**) fractions of HEK293 cells. Controls: TOMM70 (efficiency of mitochondrial isolation), TUBA1A (absence of cytosolic

*Figure 1 continued on next page*

*Figure 1 continued*

contaminations). (Bottom) Quantification of the abundance of each AURKA isoform in total or mitochondrial fractions. n = 3 independent experiments. (C) (Left) Intramitochondrial cleavage of endogenous AURKA in mitochondrial fractions of HEK293 cells transfected with control or PMPCB-specific siRNAs. (Right) Abundance of AURKA isoforms normalised against that of TOMM70 in control and PMPCB-depleted HEK293 cells. n = 3 independent experiments. (D) (Left) Localisation of ectopic AURKA-GFP in HEK293 cells by immunogold transmission electron microscopy (TEM) and (right) corresponding control condition without primary antibody. Table: number of gold beads per $\mu m^2$ of mitochondrial surface in the indicated mitochondrial subcompartments or non-mitochondrial cell surface (External). The relative abundance was then calculated by dividing the number of gold particles in each mitochondrial compartment by the number of 'External' particles. n = 20 images per condition from two independent experiments. Scale bar: 200 nm. (E) Isolation of mitochondrial soluble (S) and pellet (P) fractions by $Na_2CO_3$ extraction of mitochondrial fractions from HEK293 cells. Control cells were treated with NaCl. (F) Mitochondrial fractions from HEK293 cells digested with trypsin to degrade the OMM and blotted for endogenous AURKA. T = total lysate; ND = non digested mitochondrial fraction. To degrade the IMM and access the matrix, trypsin was combined with digitonin (+D) or Triton X-100 (+T). Controls for submitochondrial localisation: TOMM70 (OMM), CYCS (Inner Mitochondrial Space, IMS) and PMPCB (matrix). The submitochondrial compartments are indicated in the bottom part of the blot. $AURKA_{46}$, $AURKA_{43}$ and $AURKA_{38}$ are indicated by blue, yellow and red flags, respectively. A.U.: arbitrary units. Data represent means ±s.e.m. *p<0.05, **p<0.01, ***p<0.001 compared to the 'Control siRNA' condition (A), '$AURKA_{46}$' condition (B and C). $^aP$ <0.01 and $^bP$ <0.001 compared to the corresponding cleaved isoform in the 'Control siRNA' condition in (C). NS: not significant.

DOI: https://doi.org/10.7554/eLife.38111.003

The following figure supplements are available for figure 1:

**Figure supplement 1.** Identification of semi-tryptic peptides corresponding to the cleavage of AURKA in the mitochondrial matrix.
DOI: https://doi.org/10.7554/eLife.38111.004

**Figure supplement 2.** AURKA is imported into mitochondria *via* TOMM and its MTS is located at the N-terminus.
DOI: https://doi.org/10.7554/eLife.38111.005

**Figure supplement 3.** Kinase activities and mitochondrial localisation of AURKA ΔNter or mitoAURKA.
DOI: https://doi.org/10.7554/eLife.38111.006

peripheral Outer Mitochondrial Membrane (OMM) proteins, and we further combined trypsin with detergents to access the matrix. $AURKA_{38}$ showed a degradation pattern similar to the one of the matrix PMPCB protease (*Figure 1F*). Together, these data demonstrate for the first time that AURKA is localised to mitochondria at interphase and that it is imported and processed in the mitochondrial matrix.

Positively charged motifs that are located at the amino termini of mitochondrial pre-proteins often target the pre-proteins to mitochondria (*Chacinska et al., 2009*). Once there, pre-proteins interact primarily with the Translocase of Outer Mitochondrial Membrane (TOMM) complex, a multi-subunit machinery regulating the recognition and the entry of pre-proteins inside mitochondria (*Chacinska et al., 2009*). Given that AURKA enters mitochondria and it is processed in the matrix, we analysed whether it follows a canonical import route through TOMM. In this light, we searched for the physical proximity of AURKA with the subunits of the TOMM machinery by FRET/FLIM (*Padilla-Parra and Tramier, 2012*). Decreases in donor fluorescence lifetime in FRET/FLIM analyses indicated physical proximity compatible with protein-protein interactions between exogenously expressed AURKA and all the major TOMM subunits, TOMM 20, 22, 40 and 70 (*Figure 1—figure supplement 2A*). We therefore searched the N-terminus of AURKA for a Mitochondrial Targeting Sequence (MTS) signature. The first 30 or 100 amino acids of AURKA fused to GFP were not detected in mitochondria by immunofluorescence microscopy, or in mitochondrial fractions of HEK293 cells subjected to western blotting (*Figure 1—figure supplement 2B and C*). This indicates that the N-terminal of AURKA is not sufficient to shuttle GFP to mitochondria and it suggests the absence of a canonical MTS in these regions of the protein. However, an AURKA truncation mutant in which the first 30 amino acids were removed (AURKA ΔNter) did not localise to mitochondria by confocal microscopy and western blotting (*Figure 1—figure supplement 3A and B*) and did not interact with TOMM (*Figure 1—figure supplement 3C*), although catalytically active in vitro towards two AURKA substrates, histone H3 and RALA (*Lim et al., 2010*; *Kashatus et al., 2011*; *Bertolin et al., 2016*) (*Figure 1—figure supplement 3D*). It has been described that AURKA is needed for RALA localisation to mitochondria, and that the two proteins participate in a common pathway to regulate mitochondrial fission at mitosis (*Kashatus et al., 2011*). We therefore explored whether the import of AURKA inside mitochondria depends on the presence of RALA. We observed that the depletion of RALA does not inhibit the entry of AURKA into mitochondria. However, it impacts the abundance of AURKA to the same extent in total fractions and inside mitochondria.

Therefore, the mitochondrial import of AURKA is RALA-independent (*Figure 1—figure supplement 3E*).

Intriguingly, $AURKA_{38}$ was also detected in cytosolic fractions of HEK293 cells, indicating that the kinase is exported from mitochondria after processing (*Figure 2A*). To establish whether this feature is intrinsic to the AURKA MTS, we replaced the first 30 amino acids of AURKA with a well character-ised, strong MTS derived from cytochrome *c* oxidase, which is known to be recognised by TOMM and cleaved by PMPCB (*Chacinska et al., 2009*). This yields an exclusively mitochondrial AURKA (mitoAURKA), which was not exported back to the cytosol (*Figure 2B* and *Supplementary file 3C*). In addition, mitoAURKA did not localise at the centrosome when compared to normal AURKA (*Figure 2C*). The export of AURKA to the cytosol is further supported by the observation that ectop-ically expressed AURKA also shows abundant cytosolic staining (*Figure 2B and C* and *Figure 1—fig-ure supplement 2A*), which could include the fraction of $AURKA_{38}$ exported from mitochondria.

Taken together, our data show for the first time that AURKA bears an atypical MTS that is neces-sary but not sufficient for its intracellular transport to and from mitochondria.

## AURKA is enzymatically active at the mitochondria

We next explored whether the kinase activity of AURKA is involved in its transport to, and function within, mitochondria. To this end, we used a previously published AURKA FRET biosensor, which allows to track the activation of AURKA *via* its autophosphorylation on Thr288 by FRET/FLIM (*Bertolin et al., 2016*). The biosensor consists of a full-length AURKA carrying the donor FRET fluo-rophore EGFP at the N-terminus and the acceptor FRET fluorophore mCherry at the C-terminus, under the control of the minimal transcriptional regulatory region of AURKA to ensure the physiolog-ical expression of the kinase (*Bertolin et al., 2016*). The AURKA biosensor localises at mitochondria as endogenous AURKA in MCF7 cells does (*Figure 2D,E* and see *Figure 1A* for comparison), and the decrease in the lifetime of GFP revealed that it is autophosphorylated on Thr288 and activated in mitochondria. This activation is abolished by the AURKA inhibitor Alisertib (MLN8237) (*Figure 2D*), an ATP-analogue currently under clinical trials (*Görgün et al., 2010*). Intriguingly, AURKA cleavage inside mitochondria is also supported by the presence of a double band in western blots from mitochondrial fractions of MCF7 cells expressing GFP-AURKA-mCherry (*Figure 2E*). In these fractions we detected a band at ~110 kDa and corresponding to the full GFP-AURKA-mCherry biosensor construct, and a second band at ~70 kDa potentially representing the cleaved (imported) biosensor without the GFP moiety.

Given that AURKA undergoes a two-step proteolytic cleavage when entering mitochondria, its activation is likely to occur prior to the cleavage of the MTS by PMPCB as this step removes the entire N-terminus, which comprises the MTS and the FRET donor. In this light, we assessed whether a kinase-dead mutant of AURKA (AURKA Lys162Met) could shuttle to mitochondria. Indeed, AURKA Lys162Met was not retrieved in mitochondrial fractions (*Figure 2F*), indicating that the kinase activity of AURKA is required for its mitochondrial localisation.

The enzymatic activity of AURKA at mitochondria suggests that the kinase might be involved in the regulation of mitochondrial functions. To understand the potential role played by AURKA at this compartment, we explored how the over-expression of the kinase may act on two interlinked mito-chondrial functions: mitochondrial dynamics and energy production.

## AURKA regulates mitochondrial morphology

The mitochondrial dynamics balance has been shown to be crucial in cancer progression, as the mitochondrial network reshapes to meet the increasing energy requirements of cancer cells (*Vyas et al., 2016*; *Wai and Langer, 2016*). As we detected a role of over-expressed AURKA in the control of mitochondrial energy production, we next evaluated whether AURKA plays a role in mito-chondrial dynamics. First, we analysed mitochondrial morphology by TEM in HEK293 cells. Knock-down of AURKA led to mitochondrial elongation: the organelles appeared swollen, but they showed intact cristae and no apparent loss of intramitochondrial content (*Figure 3A*). Analyses of mitochon-drial length and branching (*Koopman et al., 2005*) revealed that the silencing of AURKA increases the length of the whole mitochondrial network (*Figure 3—figure supplement 4A*), confirming previ-ous results obtained with a kinase-dead version of AURKA (*Kashatus et al., 2011*). Intriguingly, TEM analyses showed that mitochondria interconnectivity increases also when AURKA is over-expressed,

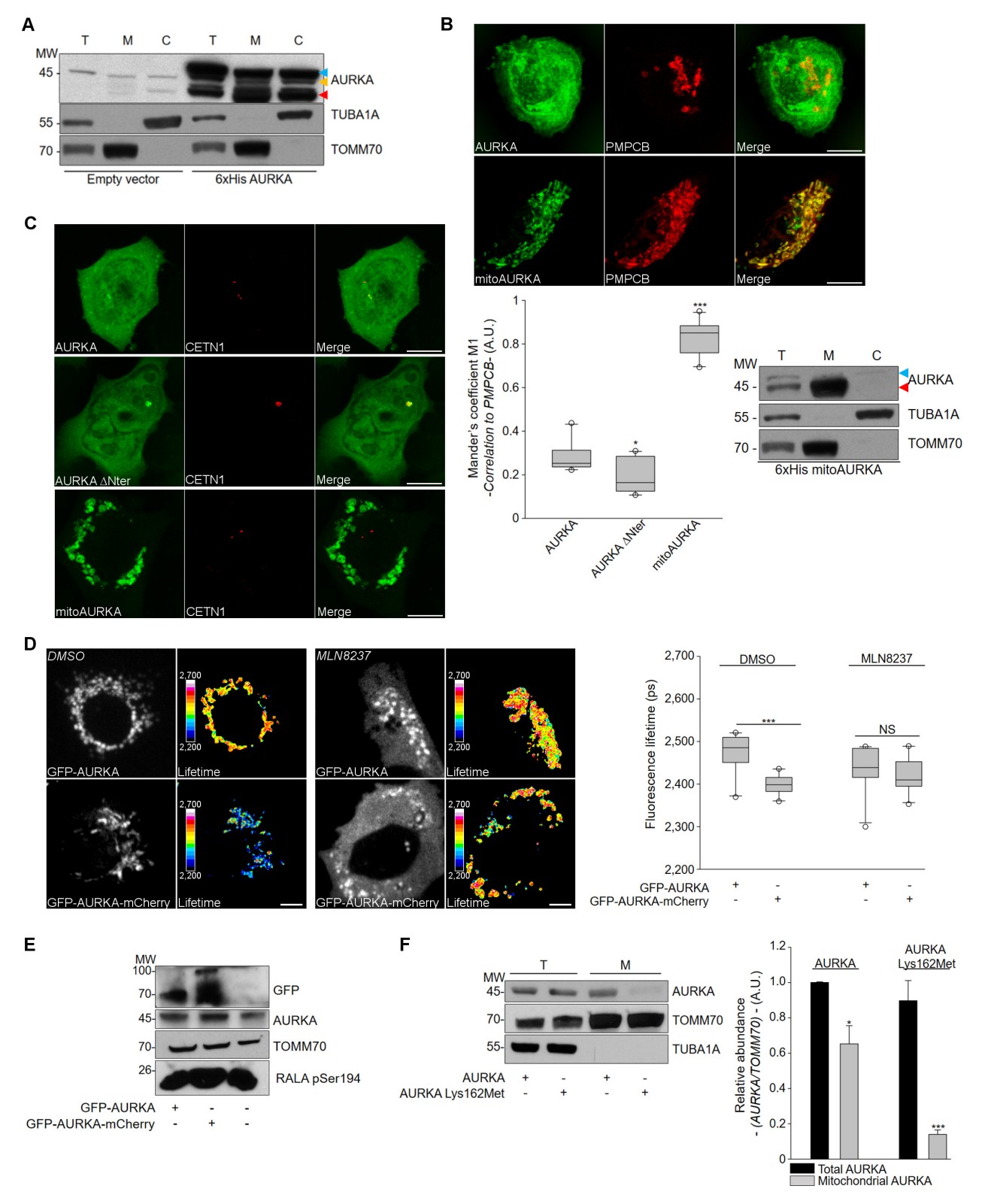

**Figure 2.** AURKA is exported to the cytosol after intramitochondrial cleavage and it is active in the mitochondrion. (**A**) Representative western blot of the intramitochondrial cleavage of endogenous and 6xHis AURKA in total (T), mitochondrial (M) and cytosolic (C) fractions from HEK293 cells. (**B**) (Top) Confocal images of MCF7 cells transfected with AURKA-GFP or mitoAURKA-GFP. Mitochondria were stained with PMPCB. (Bottom left) Mander's M1 co-localisation coefficient between AURKA-GFP, AURKA ΔNter-GFP or mitoAURKA-GFP and PMPCB. *n* = 10 cells per condition from one

*Figure 2 continued on next page*

Figure 2 continued

representative experiment (of three). (Bottom right) Mitochondrial cleavage of mitoAURKA 6xHis cDNA in total (T), mitochondrial (M) and cytosolic (C) fractions from HEK293 cells. (C) Representative confocal images of MCF7 cells transfected as indicated. CETN1-iRFP670: centrosome marker. (D) (Left) Representative fluorescence (GFP channel) and lifetime images of MCF7 cells showing the mitochondrial localisation of GFP-AURKA or GFP-AURKA-mCherry expressed under the control of the minimal AURKA promoter (*Bertolin et al., 2016*) and treated with DMSO or with the AURKA inhibitor MLN8237 (Alisertib). (Right) Corresponding ifetime quantifications. *n* = 10 cells per condition from one representative experiment (of three). Scale bar: 10 µm. (E) Representative western blot of mitochondrial fractions obtained from MCF7 cells expressing GFP-AURKA or GFP-AURKA-mCherry as in (D). (F) (Left) Abundance of AURKA and AURKA Lys162Met 6xHis cDNA and normalised to that of TOMM70 in total and mitochondrial fractions of HEK293; (right) corresponding quantification. AURKA was detected with an anti-His antibody. *n* = 3 independent experiments. Scale bar: 10 µm. AURKA$_{46}$ and AURKA$_{38}$ are indicated by blue and red flags, respectively. A.U.: arbitrary units. Data represent means ±s.e.m. *p<0.05, ***p<0.001 compared to the compared to the 'AURKA-GFP' condition (B), the corresponding 'GFP-AURKA' condition (D) or to each 'Total AURKA' condition (F). NS: not significant.

DOI: https://doi.org/10.7554/eLife.38111.007

again with no apparent signs of intramitochondrial content loss (*Figure 3A*). Under these conditions, analyses of mitochondrial network morphology showed that mitochondria are interconnected and they pack into mitochondrial clusters (*Figure 3—figure supplement 4A*).

We then compared whether the degree of mitochondrial interconnectivity when AURKA is silenced or over-expressed. To this end, we used the diffusion of a photoconvertible Dendra2 targeted to the mitochondrial matrix (mitoDendra2). This fluorescent protein is photoconverted from green to red with a 405 nm laser, and the diffusion of the red species throughout the network is achieved only if mitochondria are organised in an electrochemical continuum. In parallel, we knocked-out AURKA in a multicellular organism as the fruit fly. In contrast to tumorigenic cells having heterogeneous genotypes and where the complete depletion of AURKA is not achievable, the fruit fly allows to compare the effects of the physiological abundance of AURKA on mitochondria to the ones observed after complete knock-out or overexpression of the kinase. In this model, we measured mitochondrial connectivity in the notum, a monolayer of epithelial cells that display three-dimensional and dynamic spatial organisation of the mitochondrial network. AURKA loss-of-function mutants or those harbouring an *AURKA*-targeted RNAi and gain-of-function mutants (overexpression of *Drosophila* AURKA) showed more interconnected mitochondria than did controls (*Figure 3B*, *Figure 3—figure supplement 1B and C*, red panels). By using the diffusion of red mitoDendra2, we confirmed that mitochondria are interconnected also in MCF7 cells both when AURKA is silenced or over-expressed (*Figure 3—figure supplement 1D*).We observed that mitochondrial interconnectivity in the presence of over-expressed AURKA depends on its capacity to be imported/exported, and on the kinase activity of AURKA itself, as the over-expression of AURKA ΔNter, mitoAURKA or the kinase-dead AURKA Lys162Met have no effect on mitochondrial elongation (*Figure 3—figure supplement 2A and B*). Therefore, the role of AURKA in the regulation of mitochondrial morphology is conserved in flies and humans with no differences in mitochondrial connectivity due to the silencing or the overexpression of the kinase.

## AURKA regulates mitochondrial fission in physiological conditions

Given that an increase in mitochondrial connectivity is observed both when AURKA is up- or downregulated, we sought to define the respective molecular mechanisms involved. We first analysed the abundance of the proteins involved in mitochondrial fusion and fission when AURKA was silenced or over-expressed. When AURKA was downregulated, the abundance of MFN1 and OPA1 increased while the level of DNM1L decreased (*Figure 3C* and *Figure 3—figure supplement 3A*). We then analysed the phosphorylation state of DNM1L on Ser637, since DNM1L localises in the cytosol when phosphorylated on this residue and to mitochondrial when dephosphorylated (*Kashatus et al., 2011*; *Wai and Langer, 2016*). Downregulation of *AURKA* lead to an increase of Ser637 phosphorylation, corresponding to an increased cytosolic localisation of DNM1L (*Figure 3—figure supplement 3B*). Under these experimental conditions, increased mitochondrial connectivity at interphase could be reverted by normal AURKA expressed at physiological levels. On the contrary, the cytosolic-only AURKA ΔNter did not rescue mitochondrial elongation (*Figure 3—figure supplement 3C*), although this protein retains its catalytic activity and its capacity to phosphorylate RALA on Ser194 (*Figure 3—figure supplement 3D*).

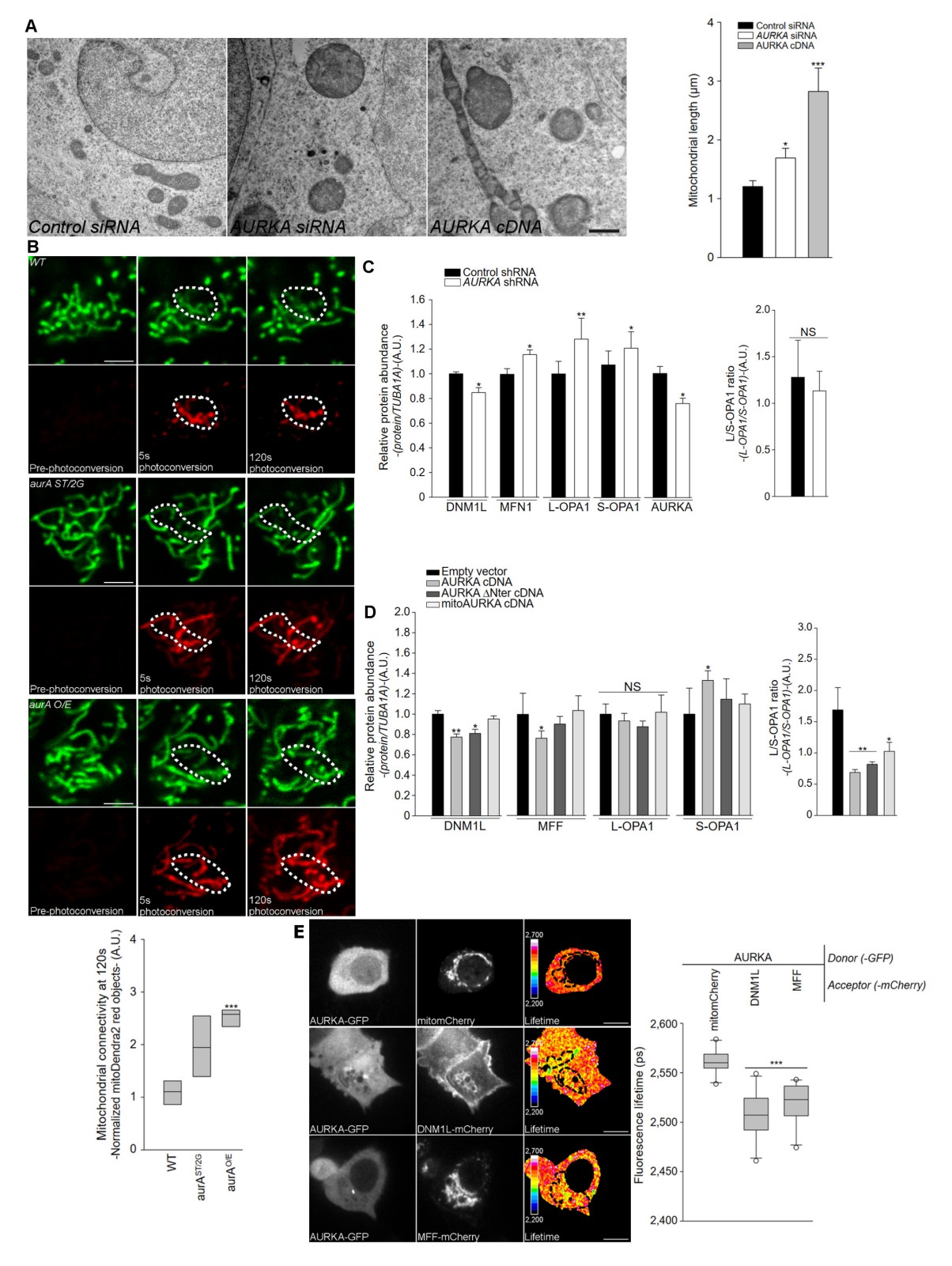

**Figure 3.** AURKA promotes mitochondrial fusion *in cellulo* and in vivo. (**A**) Mitochondrial ultrastructure of HEK293 cells by TEM transfected with control (first panel) or AURKA-specific (second panel) siRNAs, or with a cDNA encoding AURKA (third panel); quantification of mitochondrial length (fourth panel). *n* = 20 images per condition from two independent experiments. Scale bar: 200 nm. (**B**) MitoDendra2 photoconversion analysis for mitochondrial connectivity in wildtype (WT) flies, AURKA null (AurA ST/2G) and gain-of-function (AurA O/E) *Drosophila* mutants. Photoconversion area: *Figure 3 continued on next page*

Figure 3 continued

dotted line. n = 80 images per condition from eight independent pupae obtained from three independent crossings. Scale bar: 5 μm. A.U.: arbitrary units. (**C**) Quantification of the abundance of the indicated mitochondrial fusion and fission proteins of total lysates of HEK293 cells transfected with control or AURKA-specific shRNAs. (**D**) Quantification of the indicated mitochondrial proteins from total lysates of HEK293 cells transfected with an empty vector, 6xHis AURKA or the indicated variants all fused to a 6xHis tag and blotted. n = 3 independent experiments. (**E**) FRET by FLIM analysis on MCF7 cells expressing AURKA-GFP together with DNM1L-mCherry, MFF-mCherry or with a mitochondrially-targeted mCherry (mitomCherry) used as a control. Mitochondria: dotted area. (Right panels). Pseudocolour scale: pixel-by-pixel lifetime. Scale bar: 10 μm. Data represent means ±s.e.m., *p<0.05, **p<0.01, ***p<0.001 compared to the corresponding 'Control siRNA' condition (**A, C**), 'WT' genotype at 5 s after photoconversion (**B**), the 'Empty vector' condition (**D**) or 'AURKA-GFP/mitomCherry' donor-acceptor pair (**E**).

DOI: https://doi.org/10.7554/eLife.38111.008

The following figure supplements are available for figure 3:

**Figure supplement 1.** AURKA remodels the mitochondrial network by increasing mitochondrial connectivity.

DOI: https://doi.org/10.7554/eLife.38111.009

**Figure supplement 2.** Mitochondrial connectivity requires the import/export cycle of AURKA and its catalytic activity.

DOI: https://doi.org/10.7554/eLife.38111.010

**Figure supplement 3.** AURKA regulates mitochondrial fission when expressed at physiological levels while ectopic AURKA drives mitochondrial clustering and enhances mitochondrial fusion.

DOI: https://doi.org/10.7554/eLife.38111.011

The phosphorylation of DNM1L on Ser616 was previously shown to play an active role in promoting organelle fission at mitosis (*Kashatus et al., 2011*). On the contrary, the phosphorylation of DNM1L on Ser637 observed at interphase and after silencing of AURKA indicates a lack of fission, which then results in increased mitochondrial connectivity. In physiological conditions, two distinct pathways appear to regulate mitochondrial fragmentation according to the cell cycle phase. At mitosis, fission mechanisms require translocation of RALA and its effector RALBP1 to the mitochondrion (*Kashatus et al., 2011*), whereas this step is dispensable at interphase.

## AURKA regulates mitochondrial fusion when over-expressed

When AURKA was over-expressed, we observed the increase of the short isoform of OPA1 (S-OPA1), a phenomenon previously reported to be directly caused by an increased mitochondrial respiratory chain activity (*Mishra et al., 2014*). In addition, we also observed a small but significant decrease of the levels of DNM1L and its receptor MFF (*Figure 3D* and *Figure 3—figure supplement 3D*). Under these conditions, the phosphorylation of DNM1L on Ser637 remained globally unaltered (*Figure 3—figure supplement 3E*). While analysing mitochondria in tumorigenic cells, we observed that nearly 60% of cells with ectopic AURKA showed interconnected mitochondria clustered in the perinuclear region (*Figure 3—figure supplement 3F*). These structures are similar to insoluble mito-aggresomes, aggregates of mitochondria which cannot be degraded (*Driscoll and Chowdhury, 2012*). As mito-aggresomes, we found these AURKA-positive mitochondrial aggregates to be SDS-insoluble as indicated by dot-blot filter retardation assays (*Figure 3—figure supplement 3G*).

In vivo, the over-expression of the fly homologue of DNM1L – Drp1 – in flies over-expressing AURKA rescued mitochondrial interconnectivity analysed with mitoDendra2 (*Figure 3—figure supplement 3H*). Furthermore, we retrieved a direct interaction between AURKA, DNM1L and MFF by localised decrease of the lifetime of AURKA-GFP on mitochondria in FRET/FLIM analyses (*Figure 3E*). A similar decrease in GFP lifetime was not observed between over-expressed AURKA and the fusion protein MFN2 or the mitochondrial protein SNPJN2 not involved in mitochondrial dynamics (*Figure 3—figure supplement 3H*). This further corroborates the specificity of the interaction between AURKA, DNM1L and DMFF analysed by FRET/FLIM. Accordingly, the interaction between AURKA, DNM1L and MFF in the AURKA interactome was also detected by nanoLC-ESI MS/MS (*Supplementary file 2*).

Together, over-expressed AURKA directly interacts with the fission proteins DNM1L and MFF and drives mitochondrial elongation.

## AURKA regulates mitochondrial morphology in cancer cells

To validate our data in a cancer cell context, we examined the morphology of the mitochondrial network and its correlation to the levels of AURKA expression in four breast cancer cell lines. Hs578T and MDA-MB-231 cells show low expression levels of AURKA, whereas MDA-MB-468 and T47D express AURKA at higher levels (*Figure 4A*). In all these cell lines, we also retrieved an AURKA-positive signal at mitochondria by immunoblotting. When looking at the morphology of the mitochondrial network, mitochondria appeared more fragmented in Hs578T and MDA-MB-231 cells, and more elongated in MDA-MB-468 and T47D (*Figure 4B*). To correlate this phenotype with the abundance of AURKA, we inhibited the kinase with MLN8237. MLN8237 had no effect on mitochondrial length in MDA-MB-468 and T47D cells, which express high levels of AURKA and where the mitochondrial network is already dramatically interconnected per se. Conversely, MLN8237 increased mitochondrial length and branching in Hs578T and MDA-MB-231 cells, where the abundance of AURKA is low and mitochondria appear fragmented in basal conditions (*Figure 4B*). This is consistent with what observed in MCF7 cells upon the depletion of *AURKA* by siRNA.

Together, AURKA plays two opposing functions in mitochondrial dynamics according to its abundance in the cell: contributing to organelle fission when expressed under physiological conditions, and directly increasing mitochondrial fusion when over-expressed. These roles of AURKA are conserved in four carcinoma cell lines, further increasing the relevance of these results.

## Over-expressed AURKA increases the abundance of mitochondrial complex IV and up-regulates ATP production

We then analysed the mitochondrial energy capacity in HEK293 and MCF7 cells, in the presence and absence of over-expressed AURKA. We evaluated key parameters as the abundance of mitochondrial respiratory chain complexes, oxygen consumption for ATP production, the mitochondrial membrane potential and possible sources of mitochondrial stress as the activation of autophagy and of the Ubiquitin-Proteasome System (UPS). Among the levels of steady-state respiratory complexes, western blotting analysis revealed that the levels of the respiratory complex IV subunits increased in the presence of over-expressed AURKA (*Figure 5A*). It has been reported that increased abundance and activity of the respiratory complex IV are part of a mitochondrial-specific signature of epithelial cancers, which mainly rely on oxidative phosphorylation for ATP production (*Whitaker-Menezes et al., 2011*; *Vyas et al., 2016*). In this light, we observed that the oxygen consumption rate (OCR) – a measure of mitochondrial respiration – was increased when AURKA was over-expressed in HEK293 cells (*Figure 5B*) (*Whitaker-Menezes et al., 2011*; *Vyas et al., 2016*). Although only the steady-state levels of complex IV increased significantly in the presence of ectopic AURKA, the analysis of the interactome of AURKA by proteomics showed that AURKA directly interacts with multiple subunits of all respiratory complexes in HEK293 cells (*Supplementary file 2*; *Figure 5—figure supplement 1*). This reinforces the conclusion that over-expressed AURKA globally acts on the mitochondrial respiratory chain to increase ATP production.

We then analysed mitochondria-related stress levels in the presence and absence of AURKA by flow cytometry. The over-expression of AURKA increased the proportion of cells exhibiting ongoing autophagy (*Figure 5C*). On the contrary, the downregulation of AURKA increased the activity of the ubiquitin-proteasome system (*Figure 5D*), which has been proposed to be a complementary system of autophagy for the degradation of selective mitochondrial proteins (*Zhu et al., 2010*). To specifically evaluate the turnover of mitochondria, we calculated the red/green ratio of MitoTimer (*Ferree et al., 2013*). Increasing levels of red mitoTimer were observed in cells transfected with an AURKA-shRNA (*Figure 5E*), indicating that mitochondrial turnover was attenuated under these conditions. The increased red mitoTimer was specifically due to mitochondrial turnover, as we did not detect significant reactive oxygen species variations when downregulating or over-expressing AURKA (Data not shown). Mitochondrial membrane potential ($\Delta\Psi$) – another indicator of mitochondrial functionality – measured with JC-1 (*Figure 5F*) or with Tetramethylrhodamine Methyl Ester (TMRM) (*Figure 5G*) decreased upon AURKA knockdown, further confirming that mitochondria are defective in the absence of AURKA. Although mitochondria are depolarised after AURKA knockdown, no global effect on the mitochondrial oxygen consumption rate was observed under these conditions (*Figure 5H*), despite both AURKA knockdown and over-expression have an impact on cell viability as previously published (*Figure 5I*) (*Zhang et al., 2004*; *Bavetsias and Linardopoulos,*

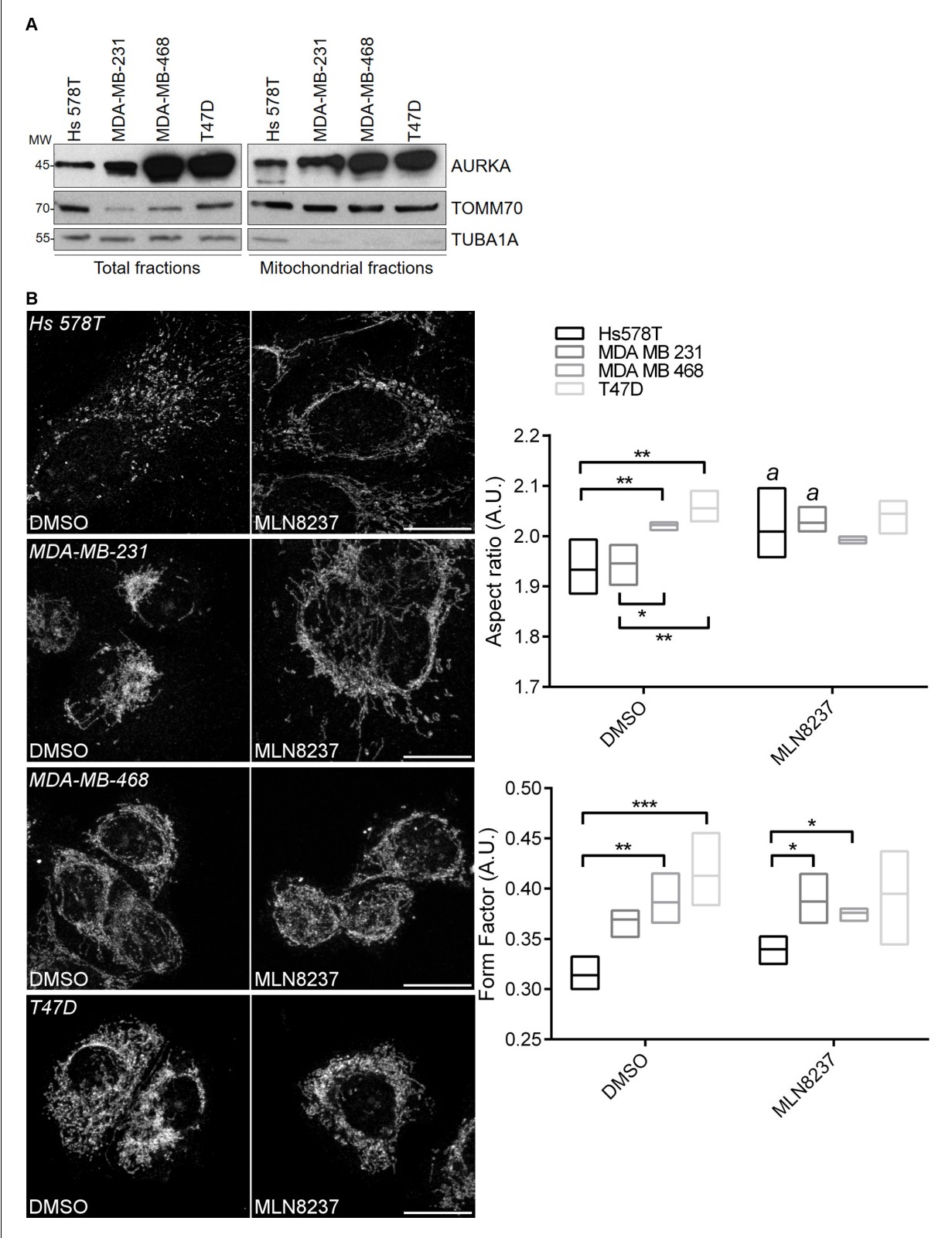

**Figure 4.** AURKA regulates mitochondrial dynamics in four carcinoma cell lines. (**A**) Representative western blots (of three) of total and mitochondrial fractions from Hs578T, MDA-MB-231, MDA-MB-468 and T47D carcinoma cells, which were probed for endogenous AURKA, TOMM70 and TUBA1A. (**B**) Representative fluorescence micrographs and corresponding quantifications of mitochondrial length (aspect ratio) and branching (form factor), and illustrating the mitochondrial network morphology in each of the four cell lines used in (**A**). Mitochondria were stained with an anti-PMPCB antibody.
*Figure 4 continued on next page*

Figure 4 continued

Where indicated, cells were treated with 100 nM MLN8237 for 3 hr prior to fixation. Scale bar: 10 μm. *p<0.05, **p<0.01, ***p<0.001 for the indicated comparisons, a = P < 0.05 compared to the corresponding 'DMSO' condition.

DOI: https://doi.org/10.7554/eLife.38111.012

2015). In addition, the mitochondrial ATP production did not differ from that of control cells when AURKA was inhibited with MLN8237, corroborating the finding that both silencing and inhibition of AURKA do not alter mitochondrial ATP production (*Figure 5L*).

In conclusion, our results indicate that AURKA maintains mitochondrial fission when expressed at physiological levels and that mitochondrial interconnectivity in the absence of AURKA is a consequence of a lack of fission. This results in the mere accumulation of elongated mitochondria without any increase in the energetic capabilities of the mitochondrial network. On the contrary, overexpressed AURKA actively enhances ATP production by promoting mitochondrial interconnectivity. These data reveal a novel role of AURKA in the control of mitochondrial bioenergetics, by acting on the mitochondrial respiratory chain and on mitochondrial functionality (*Figure 5M*).

## Discussion

Over-expression of AURKA is observed in many epithelial cancers. Increased copy number of the AURKA gene region is generally associated with an aggressive disease and poor patient survival. The AURKA gene region is located on chromosome 20, and its amplification includes the enhanced expression of additional genes (e.g. genes regulating cell cycle progression, and the most well-described AURKA interactor TPX2) (*Belt et al., 2012*; *Sillars-Hardebol et al., 2012*). In addition, the overexpression of AURKA has been linked with chromosomal instability (*Baba et al., 2009*). These events are common in different cancer types as in ovarian, pancreatic, lung and colon cancers and lead to bad prognosis. For instance, the increased copy number of AURKA is associated with the evolution of colorectal polyp into carcinoma (*Carvalho et al., 2012*). In breast cancer, the overexpression of AURKA is also linked to poor survival and it is associated with the overexpression of the human growth factor receptor 2 (HER2) and progesterone receptor (*Nadler et al., 2008*). Although epithelial cancers are non-glycolytic tumours and use the OXPHOS chain to produce ATP (*Whitaker-Menezes et al., 2011*), none of the above-mentioned studies took into account mitochondrial dysfunctions caused by or appearing in the presence overexpressed AURKA. Our study is thus pioneer in correlating for the first time this multifaceted kinase and mitochondrial physiology.

In addition to its well-characterised roles in mitosis, new functions of AURKA during interphase are regularly discovered (*Mori et al., 2009*; *Bertolin et al., 2016*; *Zheng et al., 2016*). We here demonstrated that AURKA is imported in the mitochondrial matrix. To reach this compartment, AURKA physically interacts with the TOMM complex, the major entry gate for mitochondrial proteins. Once it enters mitochondria, AURKA is cleaved in a two-step process to become a fully mature mitochondrial protein, potentially capable of interacting with multiple mitochondrial partners as the mitochondrial respiratory chain subunits. We discovered that the signal required for the import of AURKA into mitochondria is located within the first 36 amino acids of the kinase. Conventionally, MTS are incapable of shuttling to mitochondria when fused at the C-terminus of a fluorophore (*Chacinska et al., 2009*). AURKA MTS is indeed atypical, as its mitochondrial import is not blocked by the presence of a GFP at its N-terminus. In addition, the inability of this MTS to shuttle a generic GFP to mitochondria further suggest that the AURKA MTS may belong to a new class of weak mitochondrial targeting signals (*Matthews et al., 2010*), previously reported to require a specific folding conformation or post-translational modification to shuttle to mitochondria (*Karniely and Pines, 2005*). The hypothesis that centrosomal proteins play additional roles at mitochondria has already been raised (*Moore and Golden, 2009*). It has been shown that the mitochondrial protein SUCLA2, which catalyses the conversion of succinyl CoA into succinate inside mitochondria, has a mitochondrial and centrosomal double localisation in *Drosophila* (*Hughes et al., 2008*). It was shown that centrosomal SUCLA2 regulates the number and the stability of centrosomes, and this raises the fascinating hypothesis that mitochondrial proteins could in turn play roles at the centrosome under certain conditions (e.g. the cell cycle phase). Given that AURKA is preferentially a centrosomal protein now shown to directly regulate mitochondrial functions, it is tempting to speculate that this

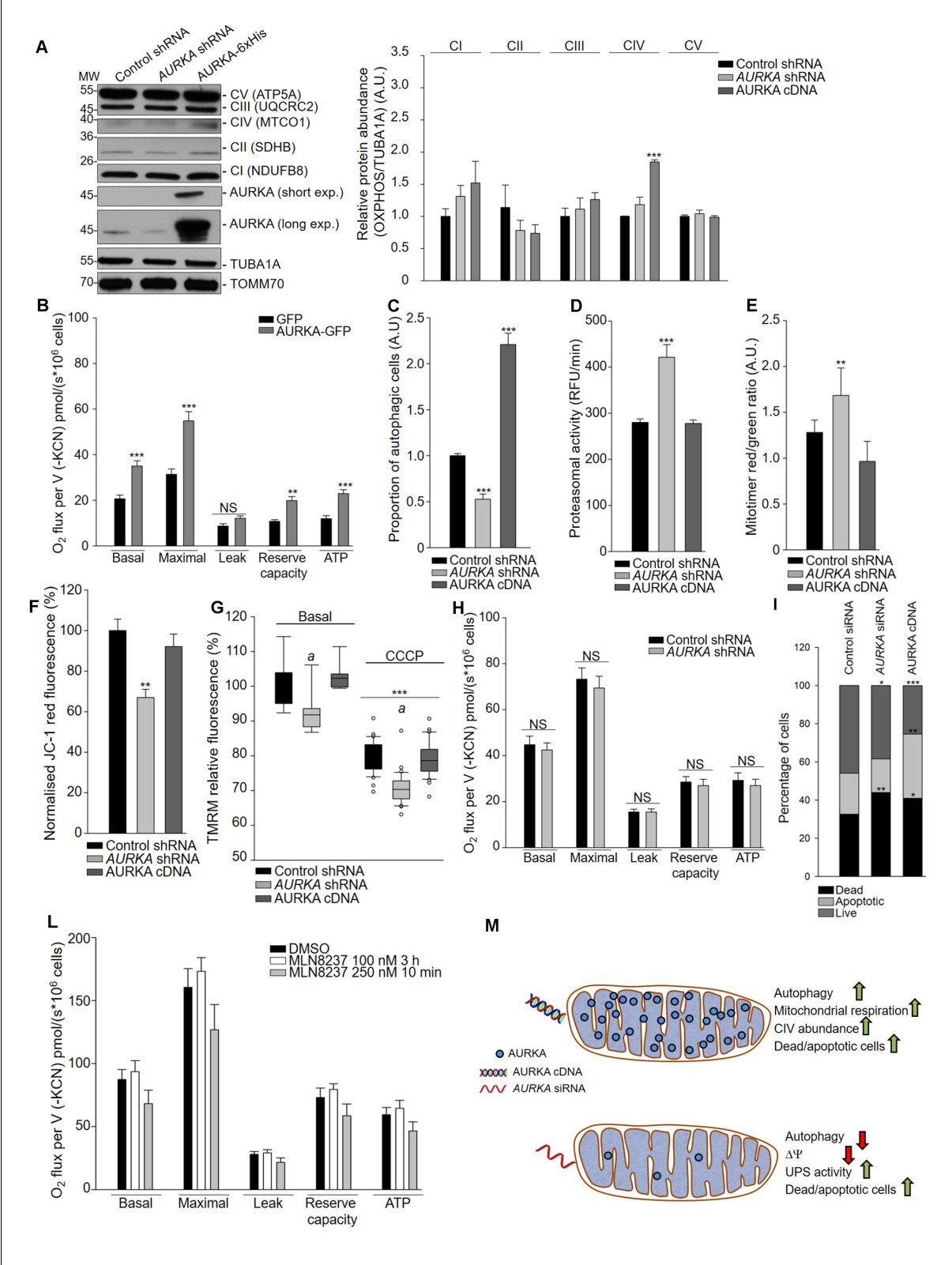

**Figure 5.** Ectopic AURKA increases mitochondrial respiration together with the abundance and the functionality of the mitochondrial respiratory chain. (A) (Top) Western blot of total lysates of HEK293 cells transfected as indicated and (bottom) corresponding quantification of the relative abundances of the indicated oxidative phosphorylation complexes subunits, representing the abundance of the five oxidative phosphorylation complexes. *n* = 3 independent experiments. (B) Mitochondrial respiration of HEK293 cells over-expressing GFP or AURKA-GFP. *n* = 3 independent experiments. (C) *Figure 5 continued on next page*

*Figure 5 continued*

Percentage of MCF7 cells over-expressing or silenced for AURKA and with activated MAP1LC3A-II and analysed by flow cytometry. $n = 3$ independent experiments. (D) Proteasomal activity in MCF7 cells transfected as indicated and analysed by flow cytometry. $n = 3$ independent experiments. RFU: relative fluorescence units. (E) MitoTimer red/green ratio in MCF7 cells transfected as indicated. $n = 3$ independent experiments. (F) Red fluorescence of the mitochondrial potentiometric probe JC-1 in MCF7 cells transfected as indicated and analysed by flow cytometry. The decrease in red JC-1 fluorescence provides a readout of $\Delta\Psi$loss. $n = 3$ independent experiments. (G) TMRM relative fluorescence of MCF7 cells transfected as indicated and treated with DMSO (basal conditions) or carbonyl cyanide m-chlorophenyl hydrazone (CCCP). $n = 3$ independent experiments. (H) Mitochondrial respiration of HEK293 cells over-expressing a control or an *AURKA*-specific shRNA. $n = 3$ independent experiments. (I) Percentage of live, apoptotic and dead MCF7 cells analysed by flow cytometry and identified by the incorporation of Annexin V. $n = 3$ independent experiments. (L) Mitochondrial respiration of HEK293 cells treated with DMSO or MLN8237 at a concentration of 100 nM for 3 hr or of 250 nM for 10 min. $n = 3$ independent experiments. (M) Cartoon diagram of AURKA silencing or overproduction acting differentially on key mitochondrial functions. Green arrows: upregulation; red arrows: downregulation. UPS: ubiquitin-proteasome system; CIV: mitochondrial complex IV. Data represent means ±s.e.m. ***$p<0.001$ compared to each corresponding 'Control shRNA' condition, (A, C–F), the 'GFP' condition (B), each 'basal' condition (G), or to each corresponding 'Control siRNA' condition (H–I). $^{a}P <0.01$ compared to the corresponding 'Control siRNA' condition for each treatment (G). All comparisons in (L) were not significant.

DOI: https://doi.org/10.7554/eLife.38111.013

The following figure supplement is available for figure 5:

**Figure supplement 1.** Mitochondria-associated proteins co-eluting with affinity-purified AURKA-GFP are over-represented.
DOI: https://doi.org/10.7554/eLife.38111.014

mitochondria-to-centrosome crosstalk could also work in a retrograde manner from the centrosome to mitochondria, with centrosomal proteins as AURKA also regulating mitochondrial functions. In this light, it will be interesting to explore whether the centrosomal and mitochondrial pools of AURKA are spatiotemporally connected. To this end, further studies are required to establish the exact molecular mechanisms that allow the first 36 amino acids of AURKA to act as a MTS only when bound to the rest of the protein.

We then searched for potential roles of AURKA at mitochondria and for impairments of mitochondrial functionality when AURKA is over-expressed. We discovered that the over-expression of AURKA enhances mitochondrial ATP production. In exploring the mitochondrial interactome of AURKA during interphase, we discovered that one out of five interactors of AURKA is a mitochondrial protein as revealed by nanoLC-ESI MS/MS analyses (*Supplementary file 2*, *Figure 5—figure supplement 1*). Proteins regulating energy metabolism in the cell, including multiple subunits of the mitochondrial respiratory chain, were found to significantly interact with AURKA at interphase (*Supplementary file 2*, *Figure 5—figure supplement 1*). This strongly supports a role of AURKA in the control of the mitochondrial respiratory chain functionality.

It is known that an interconnected mitochondrial network favours ATP production through mixing of the intramitochondrial content, which also counteracts the effects of deleterious mtDNA mutations in vivo (*Nakada et al., 2001*). Interconnected mitochondrial networks have been proposed to act as energy-transmitting cables, delivering energy to parts of the cell in which oxygen for mitochondrial respiration is low (*Westermann, 2012*). Mitochondrial fusion is also stimulated in selected stress conditions as starvation, helping the cell to cope with increasing energy demands (*Tondera et al., 2009*). The over-expression of AURKA represents a mitotic stress paradigm with centrosome abnormalities, chromosome misalignment, aberrant DNA inheritance at cell division and apoptosis (*Zhang et al., 2004*; *Nikonova et al., 2013*). Therefore, it is not surprising that mitochondria under these conditions modify their functionality beyond mitosis as well, adapting to stress by increasing their connectivity and the production of ATP during interphase. On the contrary, the increased mitochondrial connectivity observed in the absence of AURKA or when the kinase is pharmacologically inhibited does not lead to an increased ATP production. However, the connectivity of the mitochondrial network under these conditions resembles the one observed when AURKA is over-expressed. As AURKA drives mitochondrial fission when expressed at physiological levels, the absence of AURKA or the inhibition of its catalytic activity lead to the appearance of interconnected mitochondrial networks. In the absence of an active AURKA, the paradigms regulating mitochondrial fission are therefore limited and fusion remains the only mechanism left to regulate mitochondrial morphology, as previously proposed in conditions of fission inhibition (*Hoitzing et al., 2015*). In this light, we indeed demonstrate that the molecular mechanisms used by AURKA to regulate

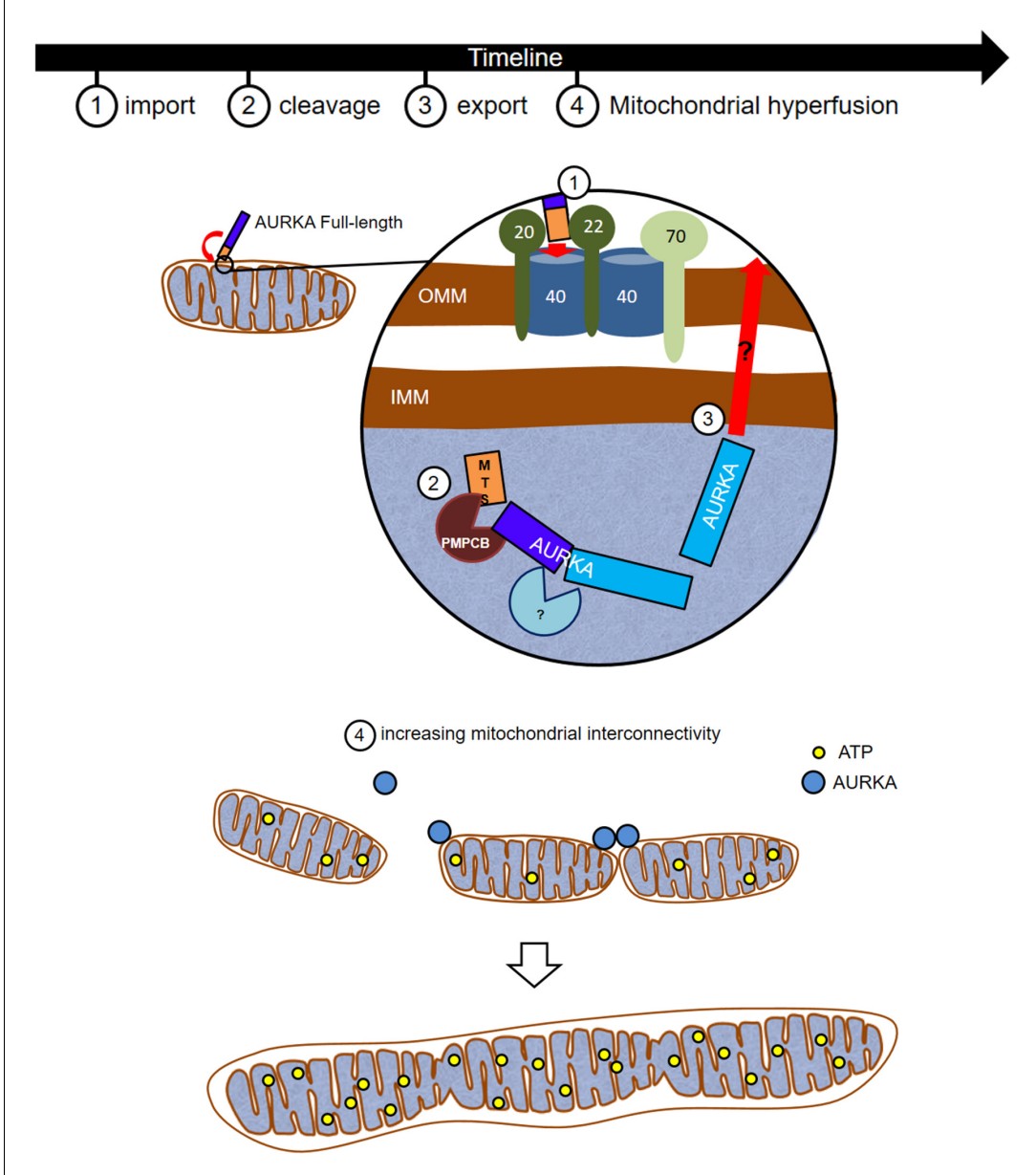

**Figure 6.** AURKA induces mitochondrial hyperfusion *via* its import/export cycle to enhance mitochondrial ATP production. (1-3) Timeline and corresponding representation of the import of AURKA via TOMM, the first proteolytic cleavage of AURKA inside the mitochondrial matrix by the PMPCB protease and the second proteolytic cleavage by a still unknown protease. Subsequent mechanisms leading to the export of AURKA in the cytosol after mitochondrial cleavage remain to be elucidated. (4) Exported AURKA induces the fusion of remaining mitochondria in an interconnected network. AURKA-dependent mitochondrial fusion leads to increased ATP levels.

DOI: https://doi.org/10.7554/eLife.38111.015

mitochondrial dynamics are distinct according to the expression levels of the kinase. When AURKA is silenced, fission is inhibited. In conditions where AURKA is overexpressed, fusion is enhanced and so is the energetic capability of the entire mitochondrial network. In conclusion, the modifications induced by AURKA to mitochondrial morphology are multifaceted and, in this context, the simple interconnectivity of the mitochondrial network is not a direct readout of the energetic capabilities of mitochondria.

The AURKA interactome and FRET by FLIM analyses show that AURKA directly interacts with DNM1L and MFF. A direct interaction of cytosolic AURKA with proteins regulating mitochondrial

dynamics is conceivable, as they are mainly located on the OMM. Nevertheless, the physical interaction of the kinase with multiple components of the mitochondrial respiratory chain, located on the Inner Mitochondrial Membrane, would require the kinase to be imported into mitochondria to ultimately promote ATP production. This is also in agreement with an increased abundance of the Inner Mitochondrial Membrane protein S-OPA1 induced by AURKA overexpression. Of note, increased S-OPA1 is a previously characterised hallmark of augmented mitochondrial respiratory chain activity (*Mishra et al., 2014*), which further reinforces our conclusion that over-expressed AURKA directly potentiates the functionality of the mitochondrial respiratory chain. However, further studies are required to understand how AURKA spatiotemporally interacts with its multiple partners and if different sub-mitochondrial pools of AURKA are capable of driving changes in mitochondrial morphology and in energy production.

Recent findings showed that mitotic AURKA promotes mitochondrial fission through the phosphorylation of RALA on Ser194 in the cytosol (*Kashatus et al., 2011*). Through this modification, RALA can shuttle to mitochondria to ensure the correct segregation of these organelles to daughter cells. We therefore sought to understand whether the role played by AURKA at mitochondria during interphase falls within the interplay between AURKA and RALA. First, the kinase is imported into mitochondria regardless of RALA. Second, the MTS of AURKA is the only portion of the protein strictly required for the pro-fission role of AURKA in physiological conditions. Indeed, a version of AURKA devoid of this fragment is unable to induce organelle fragmentation although it is fully capable of phosphorylating RALA on Ser194. Therefore, our study demonstrates that AURKA can impact on mitochondrial functions following two parallel pathways: a RALA-independent pathway during interphase, and by interacting with RALA at mitosis (*Kashatus et al., 2011*). Although we demonstrated the existence of two different pathways taken by AURKA, further studies are necessary to better characterise their molecular players and their potential interplay.

In conclusion, we propose the mitochondrial pool of AURKA regulates the fusion of interconnected organelles and, by doing so, it controls ATP production (*Figure 6*). Mitochondria with high metabolic capacity might escape turnover through fusion mechanisms and thus sustain the high metabolic needs of cancer cells, potentially representing a selective advantage for epithelial cancer progression. Targeting mitochondrial hyperfunctionality together with AURKA inhibition might therefore represent an innovative approach in the development of anti-cancer treatments.

# Materials and methods

**Key resources table**

| Reagent type (species) or resource | Designation | Source or reference | Identifiers | Additional information |
|---|---|---|---|---|
| Genetic reagent (*D. melanogaster*) | WT (w$^{1118}$) | NA | NA | |
| Genetic reagent (*D. melanogaster*) | Sca-Gal4 | PMID: 2125959 | NA | |
| Genetic reagent (*D. melanogaster*) | mitoDendra2 | Dr. Thomas Rival (University of Marseille) | NA | |
| Genetic reagent (*D. melanogaster*) | aurA ST | PMID: 23685146 | NA | |
| Genetic reagent (*D. melanogaster*) | aurA 3A | this paper | NA | aurA$^{3A}$ indel |
| Genetic reagent (*D. melanogaster*) | aurA 2G | this paper | NA | aurA$^{2G}$ indel |
| Genetic reagent (*D. melanogaster*) | aurA dsRNA | Vienna Drosophila RNAi Center | VDRC: 108446 | |
| Genetic reagent (*D. melanogaster*) | aurA exel | Bloomington Drosophila Stock Center | BDSC: 8376 | |
| Genetic reagent (*D. melanogaster*) | aurA exel | Bloomington Drosophila Stock Center | BDSC: 8377 | |

*Continued on next page*

*Continued*

| Reagent type (species) or resource | Designation | Source or reference | Identifiers | Additional information |
|---|---|---|---|---|
| Genetic reagent (*D. melanogaster*) | *Drp1* | Bloomington Drosophila Stock Center | BDSC: 51647 | |
| Cell line (H. sapiens) | MCF7 | ATCC | HTB-22 | Maintained in DMEM supplemented with 10% FBS, 1% penicillin/streptomycin and 1% L-Glutamine |
| Cell line (H. sapiens) | HEK293 | ATCC | CRL-1573 | Maintained in DMEM supplemented with 10% FBS, 1% penicillin/streptomycin and 1% L-Glutamine |
| Cell line (H. sapiens) | Hs578T | ATCC | HTB-126 | Maintained in the Legembre Lab in DMEM supplemented with 10% FBS, 1% penicillin/streptomycin and 1% L-Glutamine |
| Cell line (H. sapiens) | MDA-MB-231 | ATCC | HTB-26 | Maintained in the Legembre Lab in DMEM supplemented with 10% FBS, 1% penicillin/streptomycin and 1% L-Glutamine |
| Cell line (H. sapiens) | MDA-MB-468 | ATCC | HTB-132 | Maintained in the Legembre Lab in DMEM supplemented with 10% FBS, 1% penicillin/streptomycin and 1% L-Glutamine |
| Cell line (H. sapiens) | T47D | ATCC | HTB-133 | Maintained in the Legembre Lab in DMEM supplemented with 10% FBS, 1% penicillin/streptomycin and 1% L-Glutamine |
| Antibody | AURKA | Merck Millipore | PC742 | 1:1000 |
| Antibody | AURKA | home made | clone 5C3 (ref. 48 of this paper) | 1:20 |
| Antibody | Drp1 | BD Pharmingen | 611112 | 1:2000 |
| Antibody | Drp1 pS616 (DA91) | Cell Signaling | 4494 | 1:1000 |
| Antibody | Drp1 pS637 | Cell Signaling | 4867 | 1:1000 |
| Antibody | Fis1 (TTC1) clone EPR8412 | abcam | ab156865 | 1:1000 |
| Antibody | GFP | Roche/Sigma Aldrich | 11814460001 | 1:1000 |
| Antibody | Histone H3 pS10 | Millipore | 06–570 | 1:10,000 |
| Antibody | His tag | Covalab | mab90001-P | 1:3000 in 3% BSA |
| Antibody | Mff | abcam | ab81127 | 1:500 |
| Antibody | MitoFusin 1 | abcam | ab57602 | 1:1000 |
| Antibody | MitoFusin 2 | abcam | ab56889 | 1:500 |
| Antibody | Opa1 | abcam | ab157457 | 1:1000 |
| Antibody | OXPHOS Rodent WB Antibody Cocktail | Mitosciences | ab110413 | 1:1000 |
| Antibody | PMPCB | Proteintech | 16064–1-AP | 1:1000 |
| Antibody | RalA S194 | Millipore | 07–2119 | 1:1000 |
| Antibody | Tom22 | Abcam | ab10436 | 1:500 |
| Antibody | Tom70 | Abcam | ab106193 | 1:5000 |
| Antibody | Tubulin alpha clone YL1/2 | Millipore | MAB1864 | 1:5000 |
| Antibody | VDAC1 | Abcam | ab15895 | 1:1000 |
| Chemical compound, drug | MLN8237/Alisertib | Selleck chemicals | S1133 | |
| Chemical compound, drug | TMRM | Thermo Fischer scientific | T668 | |
| Sequence-based reagent | Allstar Control siRNA | Qiagen | SI03650318 | |

*Continued on next page*

*Continued*

| Reagent type (species) or resource | Designation | Source or reference | Identifiers | Additional information |
|---|---|---|---|---|
| Sequence-based reagent | AURKA siRNA | Eurogentec | NA | sequence 5'-AUGCCCUG UCUUACUGUCA-3' |
| Sequence-based reagent | RALA siRNA | Qiagen | SI03650318 | |
| Sequence-based reagent | PMPCB siRNA | Dharmacon | L-004747- 00–0005 | |
| Recombinant DNA reagent | AURKA shRNA | Sigma Aldrich | SHCLNG-NM_003600 | |
| Recombinant DNA reagent | Control shRNA | Sigma Aldrich | SHC002 | |

## Expression vectors and molecular cloning

Unless purchased from Addgene, DNA constructs were generated using Gibson Assembly Master Mix (New England Biolabs) and T4 DNA ligase (Thermo Fisher Scientific). All restriction enzymes were purchased from Thermo Fisher Scientific. All cloning reactions were verified on a 3130 XL sequencer (Applied Biosystems). All site-directed mutagenesis reactions were performed by Quick-Change site-directed mutagenesis (Stratagene). Vectors carrying AURKA ΔNter were constructed by removing the first 30 aminoacids of AURKA; mitoAURKA was constructed by adding the MTS of cytochrome *c* oxidase to AURKA ΔNter. The complete list of plasmid used in the study is reported in *Supplementary file 3*.

## Cell culture reagents

Mycoplasma-free MCF7 (HTB-22) and HEK293 (CRL-1573), cells were purchased from the American Type Culture Collection and grown in Dulbecco's modified Eagle's medium (DMEM, Sigma-Aldrich) supplemented with 10% foetal bovine serum (GE Healthcare), 1% L-glutamine (GE Healthcare) and 1% penicillin–streptomycin (GE Healthcare). Hs578T, MDA-MB-231, MDA-MB-468 and T47D cells were a kind gift of P. Legembre (CLCC Eugène Marquis, Rennes) and were grown in Dulbecco's modified Eagle's medium (DMEM, Sigma-Aldrich) supplemented with 10% foetal bovine serum (GE Healthcare), 1% L-glutamine (GE Healthcare) and 1% penicillin–streptomycin (GE Healthcare). For live microscopy experiments, cells were incubated in phenol red-free Leibovitz's L-15 medium (Thermo Fisher Scientific) supplemented with 20% foetal bovine serum, 1% L-glutamine and 1% penicillin–streptomycin. All live microscopy experiments were performed at 37°C in Nunc Lab-Tek II Chamber slides (Thermo Fisher Scientific). Validated siRNA against PMPCB was purchased from Dharmacon (L-004747- 00–0005), AllStars negative control (SI03650318) and validated siRNA against RALA (SI02662835) were purchased from Qiagen; the siRNA against AURKA was synthesised as previously described (*Bertolin et al., 2016*) (sequence: 5'-AUGCCCUGUCUUACUGUCA-3') and purchased from Eurogentec. The *AURKA*-specific (SHCLNG-NM_003600) and non-targeting control (SHC002) shRNAs were purchased from Sigma-Aldrich. Plasmids, siRNAs and shRNAs were transfected by the calcium phosphate method or with Lipofectamine 2000 (Thermo Fisher Scientific) according to the manufacturer's instructions. Cells were plated at 70% confluence in 96-well cell plates for plate reads, in 24-well cell plates for immunocytochemistry, or on 10 or 15 cm$^2$ petri dishes for total cell lysates and subcellular fractionation. Cells were harvested, fixed or imaged 48 hr after transfection unless otherwise indicated. MLN8237/Alisertib was purchased from Selleck Chemicals and used at a final concentration of 250 nM for 10 min or of 100 nM for 3 hr before imaging, cell fixation or harvesting. Tetramethylrhodamine methyl ester perchlorate (TMRM, Thermo Fisher Scientific, 50 nM) was incubated with the cells for 30 min at 37°C in phenol-free medium before imaging.

## Generation of AurA knockout *Drosophila* strains

*AurA*$^{2G}$ and *aurA*$^{3A}$ indels were generated using the approach described in (*Kondo and Ueda, 2013*). Briefly, gRNAs targeting exon 2 (GGCGCTTTGATCAGGAAGCCAGG) or exon 3 (GGAAAAGGAATCCCAGTTCGTGG) of AurA were cloned into the pBFv-U6.2 vector. Following molecular validation by sequencing, the pBFv-U6.2. exon2 gRNA or pBFv-U6.2. exon3 gRNA plasmids were injected into *y*$^1$ *v*$^1$ P{nos-PhiC31C\int.NLS}X; P{caryP}attP40 (stock BDSC25709) by Rainbow Transgenic Flies, Inc. The resulting male transformants *y*$^1$ *v*$^1$ P{nos-PhiC31C\int.NLS}X; P{U6.2-exon2 (or exon 3)gRNA} attP40 were balanced over CyO using *y*$^2$ *cho*$^2$ *v*$^1$; *Sp/CyO* stock. Male *y*$^2$

cho[2] v[1]; P{U6.2-exon2 (or exon 3)gRNA}attP40/CyO flies were next crossed with female y[2] cho[2] v[1]; P{nos-cas9}attP40/CyO flies. The resulting male y[2] cho[2] v[1]; P{nos-cas9}attP40/P{U6.2-exon2 (or exon 3)gRNA}attP40; aurA [(indel ?)] flies were crossed with w; kr[If]/CyO; MKRS/TM6, Tb, Hu females. Single males of the genotype w; P{nos-cas9}attP40/CyO; AurA [(indel ?)]/TM6, Tb, Hu were crossed with w; kr[If]/CyO; MKRS/TM6, Tb, Hu females. The resulting w; r[If]/CyO; AurA [(indel ?)]/TM6, Tb, Hu stock was established and characterised by sequencing. AurA[2G] is a 17 bp deletion that induces a frame-shift and eventually a STOP codon, encoding a 100-aa protein containing the first 61 aa of AurA. AurA[3A] is a 7 bp deletion encoding a 180-aa protein containing the first 177 aa of AurA. Both alleles fail to complement aurA[ST] null allele (Moon and Matsuzaki, 2013), and no AurA protein was detected by western blot. UAS-mitoDendra2 flies were established by adding the cytochrome c oxidase MTS to the Dendra2 fluorophore. The remaining strains used in this study are listed in Supplementary file 4. Drosophila melanogaster crossings were set up and grown at 25°C. All crossings and the corresponding Fig. panels are listed in Supplementary file 5; w[1118] pupae were used as wild-type controls for all experiments. Pupae were collected as white pupae, aged for 16 hr at 25°C and mounted on glass slides prior to imaging. All images collected in this study were acquired from epithelial cells of the dorsal thorax (notum) at room temperature.

## Mitochondrial isolation, sodium carbonate extraction, trypsin digestion and in vitro kinase assays

Total protein fractions were obtained by lysing cells in 50 mM Tris-HCl (pH 7.5), 150 mM NaCl, 1.5 mM MgCl$_2$, 1% Triton X-100, and 0.5 mM dithiothreitol (DTT) supplemented with 0.2 mM Na$_3$VO$_4$, 4 mg/ml NaF, 5.4 mg/ml β-glycerophosphate and protease inhibitors (Complete Cocktail, Roche) followed by centrifugation at 13,000 g for 20 min at 4°C. Isolated mitochondrial fractions were obtained by differential centrifugation as previously described (Bertolin et al., 2013) and were digested with 1 µg of trypsin (Sigma-Aldrich) per 10 µg of protein at 37°C for the indicated incubation times (Figure 1F). Digestion of the inner mitochondrial membrane was performed by adding 2 µg/µl digitonin or 3% Triton X-100 to trypsin-digested fractions. Insoluble/soluble mitochondrial protein fractions were obtained by alkaline extraction. Briefly, mitochondrial fractions were incubated on ice with 200 mM Na$_2$CO$_3$ or NaCl as a control followed by incubation of the pellet and soluble fractions in 600 mM sorbitol (Sigma-Aldrich) and 20 mM HEPES-KOH (Sigma-Aldrich, pH 7.4). Proteins were precipitated in 12% trichloroacetic acid (Sigma-Aldrich), washed three times with acetone, dried and resuspended in Laemmli sample buffer. Cytosolic fractions were obtained from mitochondria-free protein fractions using Amicon Ultra 4 ml filters for protein purification and concentration (10 kDa cutoff, Merck Millipore) according to the manufacturer's instructions. Protein purification and in vitro kinase assays were performed as described in (Bertolin et al., 2016).

## Western blotting analyses and dot-blot filter retardation assay

All protein fractions were assayed using the Bradford reagent (Bio-Rad) and then boiled in Laemmli sample buffer, resolved by SDS–PAGE, transferred onto a nitrocellulose membrane (GE Healthcare) and analysed by western blotting. Dot-blot filter retardation assays were performed in a 96-well Bio-Dot microfiltration unit (Bio-Rad) using a 0.22 µm cellulose acetate membrane (Dutscher). After treatment, the samples were resuspended in 2% SDS, loaded onto the membrane, filtered and washed twice with 0.1% SDS. The list of primary antibodies is in Supplementary file 6. Secondary horseradish-peroxidase-conjugated antibodies (anti-mouse and anti-rabbit) were purchased from Jackson ImmunoResearch Laboratories; anti-rat antibodies were purchased from Bethyl Laboratories. The membranes were incubated with commercially available (Pierce) or homemade enhanced chemiluminescence substrate as described in (Bertolin et al., 2016). Chemiluminescence signals were captured on film (CP-BU new, Agfa Healthcare), developed using CURIX 60 developer (Agfa Healthcare) and quantified with ImageJ software (NIH). The relative abundance of specific bands of interest was calculated by normalising it towards the abundance of loading controls and indicated in each graph.

## Oxygen consumption rate measurements

Cells were trypsinised, resuspended in growth medium, and placed in the respiratory chamber of an Oroboros Oxygraph-2k (WGT). Cellular respiration was determined under basal conditions in the presence of oligomycin (1 µg/ml, Sigma-Aldrich) to estimate leakage and/or in the presence of

increasing amounts (2.5–5 µM) of CCCP (Sigma-Aldrich) to obtain maximal respiration. The respiration reserve capacity was calculated by subtracting the basal respiration from the maximal respiration. The mitochondrial respiratory control corresponded to the basal/leak ratio. 'ATP' indicates the $O_2$ consumption used for ATP synthesis. Mitochondrial respiration was inhibited by the addition of 1 mM potassium cyanide (KCN, Sigma-Aldrich).

## Mass spectrometry

AURKA-GFP isolated by affinity-purification in whole cell extracts or mitochondrial fractions were resolved by SDS-PAGE using 4–12% Criterion XT Bis-Tris gradient gel (Bio-Rad) and stained with Sypro Ruby (Bio-Rad). Bands corresponding to $AURKA_{46}$, $AURKA_{43}$ and $AURKA_{38}$ were extracted from gels and processed for in-gel digestion. Alternatively, AURKA-GFP affinity-purification extracts were subjected to in-solution digestion. Peptide samples were separated by online reversed-phase (RP) nanoscale capillary liquid chromatography (nanoLC) and analyzed by electrospray mass spectrometry (ESI MS/MS). The experiments were performed with a Dionex UltiMate 3000 nanoRSLC chromatography system (Thermo Fisher Scientific/Dionex Softron GmbH) connected to a Orbitrap Fusion Tribrid mass spectrometer (Thermo Fisher Scientific) equipped with a nanoelectrospray ion source. Mass spectra data generated by the Orbitrap Fusion Tribrid instrument (*.raw files) were analyzed with Byonic version 2.12.0 (Protein Metrics, San Carlos, USA) using the optimal search parameters (mass tolerances and post-translational modifications) generated by Preview version 2.12.0 (Protein Metrics, San Carlos, USA). Precursor mass tolerance was set to 2 ppm and the fragment mass tolerance was set to 0.2 Da. Both precursor and fragments m/z measurements were recalibrated based on Preview calculations. The following mass additions were used as variable modifications: oxidation of methionine, histidine and tryptophan [+15.9949 Da], dioxidation of methionine and tryptophan [+31.9898 Da], deamidation of asparagine and glutamine [+0.9840 Da], N-terminal protein acetylation [+42.0105 Da], phosphorylation of serine and threonine [+79.9663 Da], glutamine conversion to pyroglutamate [−17.0265] and glutamate conversion to pyruglutamate [−18.0105 Da]. Carbamidomethylation of cysteines [+57.0214 Da] was set as a fixed modification. A semi-specific trypsin digestion setting allowing for N- or C-ragged peptides was specified. Byonic was used to search the MS/MS data against the Uniprot human reference proteome (71 657 entries as of May 23, 2017) complemented with a list of common contaminants maintained by Protein Metrics and concatenated with the reversed version of all sequences (decoy mode). The Byonic automatic score cutoff option was specified to maintain the false discovery rate (FDR) for peptide spectrum matches (PSMs) in the range of 0–5%. The FDR for protein identifications was set to 1%. The cellular localizations and biological functions of identified proteins were further analyzed based on information available from the Gene Ontology (GO) classification tool available in DAVID Bioinformatics Resources (https://david.ncifcrf.gov/) (*Huang et al., 2009a*; *Huang et al., 2009b*). All peptide spectrum matches corresponding to AURKA peptides identified by Byonic were extracted from the peptide dataset for the identification of putative mitochondrial processing peptidases cleavage sites into AURKA. AURKA peptides containing N- and C-ragged termini were collected and classified relative to AURKA amino acid sequence. Only high quality peptides were considered by applying a stringent p value cutoff of 0.001 (i.e. a -log(p value) of 3.0). The number of occurrences of each non-tryptic cleavage sites was calculated and plotted relative to AURKA amino acid sequence. For the CRAPome filtering, protein datasets of affinity-purified AURKA-GFP-interacting proteins generated by LC-MS/MS analysis were grouped together to establish a repertoire of putative AURKA interactors (Supplementary Material). The protein list was filtered against CRAPome (http://crapome.org/), a contaminant repository database that scores high-confidence interaction data from AP-MS experiments. A group of 30 control AP-MS experiments based on the isolation of GFP-tagged proteins by anti-GFP antibodies coupled to magnetic Dynabeads was used as a reference database to score AURKA-GFP interactions. The CRAPome primary scores (FC-A) distribution was plotted to perform statistical analysis. The upper quartile, a value that cuts off 75% of protein identifications with the lowest FC-A scores was used as a cutoff criteria. Only the top-scoring 25% (641 out of 2601 protein IDs) proteins were accepted as putative AURKA-interacting proteins and used for Gene Ontology analysis.

## Immunocytochemistry and widefield, confocal and FLIM microscopy

Cells were fixed in 4% paraformaldehyde (Sigma-Aldrich), stained using standard immunocytochemical procedures and mounted in ProLong Gold Antifade reagent (Thermo Fisher Scientific). The antibodies used were: primary monoclonal mouse anti-AURKA, 1:20 (Cremet et al., 2003); anti-GFP, 1:1000 (Sigma-Aldrich, 11814460001); polyclonal rabbit anti-PMPCB, 1:500 (Proteintech, 16064–1-AP); and secondary anti-mouse or anti-rabbit antibodies conjugated to Alexa 674 at a 1:500 dilution, Alexa 555 or 488 both at a 1:5000 dilution (Thermo Fisher Scientific). Cells displaying mitochondrial clusters were scored after viewing in a DMRXA2 microscope (Leica) equipped with a 63X oil-immersion objective (numerical aperture (NA) 1.32) and driven by MetaVue software (Molecular Devices). Multicolour images of cultured cells were acquired with a Leica SP8 inverted confocal microscope (Leica) and a 63X oil-immersion objective (NA 1.4) driven by LAS software or alternatively with a BX61WI FV-1000 confocal microscope (Olympus) driven by Olympus FV-1000 software and equipped with a 60X oil- immersion objective (NA 1.35). Multicolour images of *Drosophila* pupae were acquired with an SPE DM 5500 microscope (Leica) and a 63X oil-immersion objective (NA 1.4). The excitation and emission wavelengths for GFP/Alexa 488 were 488 and 525/50 nm, respectively; for mCherry/Alexa 555, they were 561 and 605/70 nm. GFP was used as a FRET donor in all experiments, and its decrease was measured by FLIM microscopy as in (Bertolin et al., 2016). MitoDendra2 photoconversion was performed on a region of interest (ROI) with a 405 nm laser at 0.25% power for 5 msec on an inverted Leica SP8 confocal microscope. Images of the green ($\lambda_{ex}$: 490 nm; $\lambda_{em}$: 507 nm) and red ($\lambda_{ex}$: 553 nm; $\lambda_{em}$: 573 nm) species of mitoDendra2 were then acquired using a 63X oil-immersion objective (N.A. 1.4) and a hybrid detector, driven by the LAS software. The total number of red objects present 120 s after photoconversion was normalised to the number of red objects in the ROI in the first image obtained after the photoconversion procedure (5 s). Fluorescence co-localisation was calculated with the JaCoP plugin (Bolte and Cordelières, 2006) of the ImageJ software after applying an automatic threshold mask to the confocal images; AURKA-positive mitochondria in each cell cycle phase were calculated by normalising AURKA and PMPCB-co-localising objects to the total number of AURKA-positive objects. Mitochondrial aspect ratio and form factor were calculated from confocal images as in (Koopman et al., 2005). TMRM and mito-Timer fluorescence were acquired in a FluoSTAR OMEGA plate reader (BMG Labtech) equipped with 485/520 490 nm and 540/615 nm excitation/emission filters. When overexpressing AURKA or one of its variants for confocal microscopy, non-fluorescent 6xHis-tagged AURKA was used instead of GFP-AURKA where indicated.

## Electron microscopy

For conventional electron microscopy, the cells were rinsed with 0.15 M sodium cacodylate and fixed by adding 2.5% glutaraldehyde for 1 hr. After fixation, the cells were rinsed several times with 0.15 M sodium cacodylate and post-fixed with 1.5% osmium tetroxide for 1 hr. After further rinsing, the samples were dehydrated in increasing concentrations of ethanol (50, 70, 90% and 100% v/v). The cells were gradually infiltrated with increasing concentrations of epoxy resin (30, 50, 70% v/v in ethanol) for a minimum of 3 hr per step. The samples were then incubated overnight in pure epoxy resin before continuing the infiltration procedure with a two-step incubation in 2,4,6-Tris(dimethylaminomethyl)phenol (DMP30, Sigma-Aldrich)-epoxy resin, first for 3 hr and then at 60°C for 24 hr to polymerise the samples *en bloc*. Ultra-thin sections of 80 nm were then cut from the blocks using a UCT ultramicrotome (Leica), placed on grids, and post-stained with uranyl acetate for 30 min and with lead citrate for 20 min. For immunoelectron microscopy, cells were centrifuged for 5 min at 800x*g*, recovered and rapidly fixed in 4% paraformaldehyde and 0.1% glutaraldehyde in 0.1 M phosphate buffer (PB) for 4 hr as previously described (Slot and Geuze, 2007). The cells were rinsed in PB and suspended in gelatin (12% wt/vol) at 37°C for 10 min. After solidification on ice, the cell blocks were cut and immersed in 2.3 M sucrose at 4°C overnight. The blocks were then mounted on a pin holder and placed in a UC7 cryo-ultramicrotome (Leica). Rapid trimming was performed using a 90°C trim tool (DTB20, Diatome AG) at −80°C to determine a region of interest. Ultrathin cryosections (70–90 nm) were cut at −120°C using a dry diamond knife (DCIMM 3520, Diatome AG), picked up with a mixture (1:1 vol/vol) of 2.3 M sucrose and 2% wt/vol methylcellulose and transferred to formvar-coated copper or nickel grids. The grids were subjected to standard immunolabelling procedures (Slot and Geuze, 2007; Griffiths et al., 1984; Nicolle et al., 2015) before a final contrast on ice

with a mix of 2% wt/vol methylcellulose and 4% wt/vol uranyl acetate in a ratio of 8:2. The combinations of primary and secondary antibodies used are listed in *Supplementary file 7*. The grids used for electron microscopy were examined at 120 kV with a JEOL 1400 (Peabody) transmission electron microscope equipped with an SC 1000 camera (Gatan Orius). Mitochondrial length, lysosomal abundance and number of gold beads were scored using the ImageJ software.

### Flow cytometry

Analyses of autophagy, apoptosis, mitochondrial membrane potential and proteasome peptidase activity were performed on a BD Accuri C6 flow cytometer (BD Biosciences). Annexin V-FITC/PI apoptosis detection kit was used as described by the manufacturer (Thermo Fisher Scientific). MAP1LC3A activation was measured using the FlowCellect Autophagy LC3 Antibody-based Assay Kit (Merck Millipore). Mitochondrial inner membrane potential was measured with the JC-1 probe (Thermo Fisher Scientific) as previously described (*Agier et al., 2012*). The peptidase activity of proteasomes was monitored using the fluorogenic peptide succinyl-Leu-Leu-Val-Tyr-7-amido-4-methylcoumarin, LLVY-AMC (Sigma-Aldrich) (*Bulteau et al., 2006*).

### Statistical analyses

Two-way ANOVA tests were employed to compare two variables among multiple conditions, and one-way ANOVA when just one variables needed to be tested among multiple conditions. Student's t-test was employed to compare two conditions. Statistical tests were performed after testing data for normality. Two-way ANOVA and the Holm-Sidak method were used to compare the the effect of siRNAs and AURKA isoforms on the relative mitochondrial abundance of AURKA (*Figure 1C*, *Figure 1—figure supplement 3E*), the effect of the pharmacological treatment and the mitochondrial respiratory parameter on mitochondrial respiration (*Figure 5L*) the effect of pharmacological treatment and the fluorescence protein on lifetime (*Figure 2D*), the effect of pharmacological treatment and transfection conditions on TMRM fluorescence (*Figure 5G*), the effect of time and transfection conditions or *Drosophila* genotypes on the number of mitoDendra2 red objects (*Figure 3B*, *Figure 3—figure supplements 4B–D* and *5*) and the effect of the pharmacological treatment and the cell line on mitochondrial aspect ratio and form factor (*Figure 4B*). One-way ANOVA and the Holm-Sidak method were used to compare the relative mitochondrial abundance of AURKA isoforms (*Figure 1B*), the number of AURKA-positive mitochondria (*Figure 1—figure supplement 1A*), the effect of acceptors on FRET efficiencies for given donor-acceptor pairs (*Figures 3E* and *4F*, *Figure 1—figure supplements 2A*, *3C* and *6I*), Mander's co-localisation coefficients (*Figure 2B* and *Figure 1—figure supplement 2B*), the relative total or mitochondrial abundance of AURKA with normal or kinase-dead AURKA (*Figure 2F*), the relative abundance of each oxidative phosphorylation complex (*Figure 5A*), the proportion of autophagic cells (*Figure 5C*), the percentage of cells showing mitochondrial aggregates (*Figure 3—figure supplement 6E*), the abundance of mitochondrial fusion and fission proteins (*Figure 3D*), proteasomal activity (*Figure 5D*), MitoTimer red/green ratio (*Figure 5E*), JC-1 red fluorescence (*Figure 5F*) and the percentage of live, dead or apoptotic cells (*Figure 5I*). One-way ANOVA on ranks and Dunn's method were used to compare mitochondrial length (*Figure 3A*). One-way ANOVA on ranks and the Kruskal-Wallis method were used to compare mitochondrial aspect ratio and form factor (*Figure 3—figure supplement 4A*) and the abundance of phosphorylated DNM1L forms and their ratios to total DNM1L (*Figure 3—figure supplement 6E*). Student's t-test was used to compare Mander's co-localisation coefficients (*Figure 1A*), the relative total or mitochondrial abundance of AURKA isoforms (*Figure 1B*, *Figure 3—figure supplement 3E*), the relative $O_2$ flux (*Figure 5B and H*), *PMPCB* downregulation efficiency (*Figure 1C*), mitochondrial aspect ratio and form factor (*Figure 3—figure supplement 6C*), the abundance of mitochondrial fusion and fission proteins (*Figure 3C*), *AURKA* downregulation efficiency (*Figure 3C*) and the abundance of phosphorylated DNM1L forms and the ratios of these forms to total DNM1L (*Figure 3—figure supplement 6B*). Alpha for statistical tests used in this study was equal to 0.05.

## Acknowledgments

We are grateful to S Prigent for valuable help in macro design for the analysis of mitoDendra2, S Dutertre and C Chevalier of the Microscopy-Rennes Imaging Center (Biologie, Santé, Innovation

Technologique, BIOSIT, Rennes, France), L Deleurme of the flow cytometry and cell sorting platform (BIOSIT, Rennes, France) for assistance, F Matsuzaki (RIKEN Center for Developmental Biology, Japan) for sharing the aurA[ST] flies and T Rival (IBDM, France) for the precious gift of UAS-Mito Dendra2 strain prior to publication, P Legembre (CLCC Eugène Marquis, Rennes, France) for sharing Hs578T, MDA-MB-231, MDA-MB-468 and T47D cells, Y Gautier for valuable help in graphical art together with D Fairbrass, L Buhlman and E. Watrin for critical reading of the manuscript, S Huet for sharing its expertise on mitoDendra2 photoconversion, and C. Chapuis for technical help. Stocks obtained from the Bloomington Drosophila Stock Center (NIH P40OD018537) and from the Vienna Drosophila Research Center (VDRC) were used in this study, together with Exelixis and (*Kondo and Ueda, 2013*) for aurA[O/E] and dsRNA fly strains. This work was supported by Comité Nationale de la Recherche Scientifique, the Agence Nationale de la Recherche (ANR-11-BSV5-0023 KinBioFRET to CP, R LB and MT), by the Ligue Contre le Cancer Comité d'Ille et Vilaine, Comité du Maine et Loire et Comité de la Sarthe to MT The Proteomics Platform of the Quebec Genomics Center provided mass spectrometry analyses. GGP holds a tier 1 Canada chair in proteomics. This work was partially supported by the Canadian Institute of Health Research (CIHR). GB was supported by a fellowship from Fondation ARC pour la Recherche contre le Cancer and by a fellowship from Fondation Tourre pour la recherche fondamentale contre le cancer.

## Additional information

### Funding

| Funder | Grant reference number | Author |
|---|---|---|
| Association pour la Recherche sur le Cancer | post-doc grant | Giulia Bertolin |
| Ligue Contre le Cancer | Grand ouest grant | Marc Tramier |
| Fondation Tourre | post-doc grant | Giulia Bertolin |
| Agence Nationale de la Recherche | KinBioFRET | Roland Le Borgne Claude Prigent Marc Tramier |

The funders had no role in study design, data collection and interpretation, or the decision to submit the work for publication.

### Author contributions

Giulia Bertolin, Conceptualization, Data curation, Formal analysis, Investigation, Methodology, Writing—original draft, Project administration; Anne-Laure Bulteau, Marie-Clotilde Alves-Guerra, Agnes Burel, Investigation, Methodology; Marie-Thérèse Lavault, Stephanie Le Bras, Guy G Poirier, Methodology; Olivia Gavard, Resources; Jean-Philippe Gagné, Data curation, Formal analysis, Investigation, Methodology; Roland Le Borgne, Resources, Methodology, Writing—review and editing; Claude Prigent, Resources, Writing—review and editing; Marc Tramier, Supervision, Funding acquisition, Writing—review and editing

### Author ORCIDs

Giulia Bertolin http://orcid.org/0000-0002-7359-5733
Roland Le Borgne http://orcid.org/0000-0001-6892-278X
Claude Prigent http://orcid.org/0000-0001-8515-8699
Marc Tramier http://orcid.org/0000-0001-8200-6446

### Decision letter and Author response

Decision letter https://doi.org/10.7554/eLife.38111.027
Author response https://doi.org/10.7554/eLife.38111.028

# Additional files

## Supplementary files

• Supplementary file 1. Identification of the semi-tryptic peptides of AURKA. The cleavage sites identified by the analysis of AURKA N- and C-ragged semi-tryptic peptides of human AURKA are indicated. Cleavage sites shown in grey derive from peptides containing aromatic residues (Phe, Tyr, Trp) that spontaneously give rise to truncated semi-tryptic fragments during MS/MS analysis. The cleavage sites shown in red originate from semi-tryptic peptides, devoid of aromatic residues, which are potentially generated by the action of mitochondrial proteases.
DOI: https://doi.org/10.7554/eLife.38111.016

• Supplementary file 2. Identification of the AURKA interactome at interphase. We report on different pages: the protein UNIPROT IDs, the common biological contaminants obtained after the CRA-Pome analysis, the proteins with a known mitochondrial localization found among the AURKA interactome; the Gene Ontology (GO) cellular components and biological processes for the AURKA-interacting proteins and the known AURKA interactors found in this analysis.
DOI: https://doi.org/10.7554/eLife.38111.017

• Supplementary file 3. Plasmid vectors used in this study. This file includes the source of the plasmids, eventual cloning sites (when applicable) and primers used for site-directed mutagenesis.
DOI: https://doi.org/10.7554/eLife.38111.018

• Supplementary file 4. *Drosophila* strains used in this study. This file includes the name, the genotype and the source/identifier of the *Drosophila* strains used.
DOI: https://doi.org/10.7554/eLife.38111.019

• Supplementary file 5. *Drosophila* crossings. This file includes the genotype of the Drosophila crossings used in this study, together with the corresponding figure panels.
DOI: https://doi.org/10.7554/eLife.38111.020

• Supplementary file 6. Primary antibodies used for western blotting. This file includes the primary antibodies used in this study together with the brand name, the catalogue number and the dilution used.
DOI: https://doi.org/10.7554/eLife.38111.021

• Supplementary file 7. Primary and secondary antibodies used for electron microscopy. This file includes the primary and secondary antibodies used together with the brand name, the catalogue number and the dilution used.
DOI: https://doi.org/10.7554/eLife.38111.022

• Transparent reporting form
DOI: https://doi.org/10.7554/eLife.38111.023

## Data availability

All data generated or analysed during this study are included in the manuscript and supporting files.

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
