## [Decision Letter]

[Editors’ note: a previous version of this study was rejected after peer review, but the authors submitted for reconsideration. The first decision letter after peer review is shown below.]

Thank you for submitting your work entitled "Aurora kinase A localises to mitochondria and modifies mitophagy and energy production when over-expressed" for consideration by *eLife*. Your article has been reviewed by three peer reviewers, including Jody Rosenblatt as the Reviewing Editor and Reviewer #1, and the evaluation has been overseen by a Senior Editor.

Our decision has been reached after consultation between the reviewers. Based on these discussions and the individual reviews below, we regret to inform you that your work will not be considered further for publication in *eLife*.

As you will see, while the reviewers found the subject interesting, they found the study to be preliminary, making its interpretation confusing. The main problem we had was there was a lack of mechanistic insight about what substrate Aurora kinase A (AURKA) would phosphorylate to alter mitochondria function and turnover. Does it require RALA/DNM1L, NM1L, or an independent mechanism? Moreover, there was concern about the AurA localization at mitochondria, which changes in different pictures in the manuscript and is not supported in other studies cited below (and in a recent BioRxiv pre-print, which we do not use to compare but mention here for your interest), something that live imaging may better address. Without a clear understanding of where AurA dynamically localizes and what it signals to in mitochondria, it is hard to clearly determine whether it has a primary role in mitochondria fission or secondary function from its role in mitosis. Unfortunately, at present, the model proposed is speculative and not sufficiently supported by the data.

Reviewer #1:

The manuscript by Bertolin et al. demonstrates a new role for AURKA in controlling mitochondrial activity and mitophagy during interphase. While previous roles for AURKA have focused predominantly on its role in mitosis regulation, this paper represents an important departure indicating its role in controlling mitochondrial form and function. They show that AURKA gets imported and cleaved at mitochondria and its loss through knockdown disrupted mitochondrial turnover and function, leading to longer mitochondria. Because increased AURKA and an activating mutation are frequent in cancer, they next examined how overexpression or expression of the mutant affects mitochondria and found that it promotes increased respiration, and fusion, while reducing the overall total mass of cellular mitochondria. This study is very interesting suggesting a new role for AURKA on mitochondrial function and turnover, which may have important implications for its overactivation in cancer cells.

1) The study is overall compelling and well written, with helpful diagrams. One point that is confusing is the mechanism by which AURKA may function. This may go beyond the depth of this study, however, it makes the findings that both overexpression and loss of AURKA lead to longer mitochondria somewhat confusing. It would be useful (at least in the Discussion) to address how both loss and gain of function could lead to similar phenotypes. Does this mean that normally, AURKA is controlling mitochondrial quality by degrading non-functional mitochondria but when overexpressed it increases fusion of functional mitochondria. This point is interesting but confusing.

a) With respect to this point, what does this sentence mean? "These results suggests that AURKA maintains mitochondrial fission when expressed at physiological levels, and in its absence a compensatory fusion could be established as a consequence of lack of fission."

2) Since there is much discussion about the role of increased AURKA function in cancer, they should provide more data regarding the prevalence of AURKA overexpression or mutations in different cancers. How common is it? Is it an indicator of poor prognosis or restricted to certain types of cancers? Do you see similar mitochondrial hallmarks in these cancers or do other mutations alter the phenotypes?

*Reviewer #2:*

This paper reports the association of AURKA, which is often amplified in particular types of cancer, with mitochondrial. The authors report that AURKA is imported into the mitochondrial matrix via the TOMM complex and is cleaved by processing proteases in the mitochondria. The authors claim that AURKA is active in the mitochondrial matrix and alters a host of mitochondrial parameters with either depleted or overexpressed. The author's data is interpreted to indicate "functional", physical and direct interactions with a variety of diverse proteins involved in mitochondrial dynamics and electron transport chain function. AURKA within the mitochondria is also associated with alterations in rates of mitophagy.

In my opinion, the data described here is not really believable nor is its physiological significance clearly demonstrated. The entire story rests on there being a pool of AURKA that is imported into mitochondria. Even this conclusion seems dubious. AURKA isn't found in either Mitocarta or Mitominer, suggesting that dozens of experiments integrated to make these databases have failed to identify AURKA in this compartment. Although there could of course be proteins that are mitochondrial that have been missed as being mitochondrial, the absence of AURKA in combination with the absence of a clear mitochondrial targeting sequence in its N-terminus raises questions.

One must then wonder how the authors either see signals for AURKA overlapping with mitochondria or the presence of processed AURKA in extracts, with the processed form diminished upon depletion of the supposed mitochondrial protease PMPCB. One possibility for the processing seen in extracts is that cell lysis results in release of the protease from the mitochondria that has the ability to cleave AURKA, a possibility that wasn't tested. The import model is seemingly made even more unlikely in that they claim that a GFP-AURKA-Cherry fusion is also recognized and imported into mitochondria. It is hard to see, given how the translocon and mitochondrial target sequences work, that this construct would be successfully imported into mitochondria and cleaved by PMPCB.

The authors perform essentially every mitochondrial assay available to them to provide evidence of effects on mitochondrial function, either by depletion of AURKA or overexpression (at leaves that appear to be ~100-fold or more higher than endogenous). Many of these assays end up being descriptive, and they paint a bewilderingly complex interplay between AURKA and mitochondria, which makes it hard to put any specific finding into a logical biological process. As an example, the authors employ immunoprecipitation of overexpressed AURKA-GFP followed by mass spectrometry and identify 2600 proteins (approximately 25% of the expressed proteome in these cells), including 449 mitochondrial proteins. They apparently have no negative controls and try to make the conclusion that because certain mitochondrial proteins are detected, that they are likely real and direct. It is well known that the vast majority of proteins present on anti-GFP resin are non-specific binders, but they haven't tried to control for this. So ultimately, they do not rigorously demonstrate that the proposed interactions with proteins that are both in the inner membrane and function at the outer membrane during fission are specific or not. Thus, it becomes very cloudy in terms of how AURKA is working. Is its main function(s) within the matrix or in the cytoplasm in the context of fission with DNM1L/MFF?

For the vast majority of experiments, the authors use either a single siRNA or a single shRNA. Thus, there are no experiments to demonstrate the absence of off target effects for any individual RNAi reagent. Given that many of the effects are small, it is conceivable that some of the conclusions drawn are based on off-target effects.

Additionally, in Figure 3 the authors conclude that high levels of AURKA lead to increases in Complex IV, but no substantial changes in other electron transport chain complexes. The weird thing here is that siRNA against AURKA also lead to slight increases in Complex IV levels, which doesn't make sense. One would think there would be opposite activities. The physiological significance of most of the measurements in this figure isn't clear.

Many of the conclusions just don't make sense and could result from overexpression effects. For example, they state "Last, no LAMP1 staining was observed in cells over-expressing AURKA ΔNter (Figure 6C), strongly suggesting that the mitochondrial pool of AURKA is responsible for mitophagy activation." It makes no sense that overexpression would completely remove LAMP1 staining in cells. Does this suggest that there are no lysosomes, for example? The authors do not look any further than this but make the conclusion that AURKA regulates autophagy – a complete non-sequiter.

Finally, the authors try to link AURKA to mitophagy. They claim they use accepted assays but this really isn't the case. The TEM assay is very difficult to quantify, and overall they are looking at very small numbers of cells and trying to make conclusions. It isn't clear why they didn't use either the accepted mito_keima flux assay or alternatively use anti-DNA immunofluorescence, which allows quantification of the amount of mitochondria based on the mitochondrial DNA abundance. Both of these assays are highly used in the field.

Based on these and many other issues with the paper, I cannot support publication in *eLife*. I think a much more rigorous analysis would need to be done. Moreover, if AURKA is functioning as a kinase in the mitochondria, what are its substrates and how does phosphorylation by AURKA alter their functions.

Reviewer #3:

In this manuscript, Aurora A is proposed to play a critical role in regulating mitochondrial fusion, mitophagy and metabolic reprogramming in cancer cells. Aurora A localizes to the mitochondria during interphase, it is active as a kinase in this organelle, and its depletion results in significant changes in mitochondria morphology. These effects seem to be mediated by changes in DNML1 activity. The Phe31Ile variant of AURKA, a mutant found in cancer patients and previously described as a tumour-susceptibility factor, displays increased mitochondrial import and ATP production, and results in and accelerated mitophagy. Altogether, these results suggest that AURKA over-expression can promote metabolic reprogramming by inducing specific changes in the mitochondrial network.

In general, the data showing mitochondrial localization of Aurora A are solid and highly interesting and it is also clear that AurkA overexpression induces significant morphological and functional changes in mitochondria. Major remaining question is whether these changes are a consequence of the a) AurkA-NML1 interaction, b) the established RALA-DNML1-dependent pathway that controls mitochondrial fission/fusion or c) a fusion/fission-independent pathway as suggested by the Phe31Ile mutant. Some of these questions are acknowledged in the Discussion but it seems that the information in the manuscript remains a bit preliminary until some of these questions are addressed more in detail.

1) The localization of AurA seems to be variable in the different images shown. For instance is only cytoplasmic in Figure 1 (or Figure 2D) with certain spotted (mitochondria) pattern, but is both cytoplasmic and nuclear in Figure 2B, C with no evidences of spotted (mitochondrial pattern). One possibility is that cell in Figure 1 is G1 whereas it might be S/G2 in Figure 2. The authors should clarify the pattern of Aurora A localization (and co-localization with mitochondria) in the different cell cycle stages.

2) It is somehow surprising that OCR does not change after Aurora A knockdown. I wonder whether the knockdown levels are not sufficient. The authors should also use AurkA inhibitors to test mitochondrial activity (Seahorse).

3) On a related issue, how did the authors manage to prevent secondary consequences of cell cycle arrest/defects in these assays? Does changes in mitochondria induced by AurkA occur in parallel to changes in cell cycle phases (entry into G2 etc.)?

4) It is not clear whether the authors are proposing a direct activity of Aurora A on DNML1-Ser637? Or changes in Ser 637 are RALA-dependent following the already established pathway? A direct interaction between AurkA and DNML1 is suggested by the proteomics data but the authors do not evaluate this functionally. One would assume that is a RALA-dependent process but his has not been tested and RALA apparently was not present in these complexes. This mechanism should be analyzed in detail and properly clarified. Perhaps rescue assays with RalA, DNML1 would help.

5) Data with the Phe31Ile mutant are quite interesting and this mutant has a significant effect in mitophagy and mitochondrial activity. However, given the lack of specific mechanism as discussed in the previous point, it is not completely clear why this mutant induces these effects. The authors indicate that the mutations favors entry into the mitochondria; however, no changes in interconnectivity are shown. As mitophagy was previously proposed to be triggered or favored by these changes in connectivity, the actual reason for the phenotype induced by the AurkA mutant is not really clear. In addition, these data actually argue against the DNML1-fusion mechanism proposed in the earlier figures in the manuscript.

[Editors’ note: what now follows is the decision letter after the authors submitted for further consideration.]

Thank you for resubmitting your work entitled "Aurora kinase A dynamically localises to mitochondria to control mitophagy and energy production" for further consideration at *eLife*. Your revised article has been favorably evaluated by Andrea Musacchio (Senior Editor), a Reviewing Editor, and three reviewers.

On the whole, there was consensus that the paper was much improved. However, some remaining issues were identified that need to be addressed before acceptance, as outlined below:

All three reviewers found the data on mitophagy not particularly convincing. There was also concern that the cancer data was not as well-developed as it might be with respect to rescuing the mutation and imaging. Therefore, we suggest that you also omit this part of the story unless you feel that it can be better supported. These are our only requests required for acceptance. I include the comments of the reviewers below only for your own reference and feedback.

We look forward to receiving a revised version of the manuscript that addresses these final issues.

*Reviewer #1:*

In the revised version of the manuscript Bertoli et al. have added a bunch of interesting results that help to delineate the possible role of Aurora A at mitochondria. In particular, the finding that Aurora A needs to be at the mitochondria and this localization is RALA independent makes an important observation after the Kashatus 2011 manuscript. Some other controls and complementary techniques (cell cycle analysis, mitotracker, etc.) added also help to reach more solid conclusions.

The only comment I have is that the characterization of the Ph31Ile mutant is a bit preliminary to make so strong conclusions. Yet, this is a very important point in the manuscript (demonstrating that a cancer-associated mutation in Aurora A functions by modulating mitochondria rather than cell cycle functions). I would suggest to testing whether the effect of the Phe31Ile mutant can be rescued by a second mutation that prevents the localization at the mitochondria.

Reviewer #2:

I was asked by *eLife* to comment on the manuscript by Bertolin et al. after the first round of reviewing. So I had a chance to evaluate both the paper and the answers to reviewers remarks. This manuscript reports on the essential findings, which was quite surprisingly overlooked before by many researches working on this kinase, that AURKA is localizes to mitochondria and is important for mitochondria functioning and quality control. To demonstrate this new role of AURKA, authors apply various approaches, including extensive microscopy analysis and proteomics, to study the consequences of AURKA depletion or overexpression on mitochondria fate, both in "normal" and cancer cells. I think that authors provided convincing answers to the concerns raised by reviewers and provided a new sets of experiments, which improve their initial observation. Nevertheless, I have a couple of remarks, that would be useful to address.

1) I am not quite convinced by the experiments that describe AURKA localization at different phases of the cell cycle (Figure 1A and Figure 1—figure supplement 1A). It does not look like the cells were synchronized at different phases. There is no description of any time-lapse experiments either. So either the AURKA localization through the cell cycle has to be studied more thoroughly or authors delete their statement (and corresponding figures) from the current version of the manuscript.

2) Although many concerns of the reviewer #2 were properly addressed, one of them, concerning mitophagy, still looks not fully proved. The point is that the majority of the experiments are set-up to follow general autophagy, but not mitophagy. I am confused by the new experiment added (Supplementary Figure 7B), which reports the fluorescence of Lysosomal marker, LysoTracker Red. What it is supposed to demonstrate? Using experiment with *Drosophila* cells is also confusing. Why *Drosophila* suddenly? What it is about *Drosophila* that cannot be done in HEK 293? (see subsection “Over-expressed AURKA triggers mitophagy of fragmented mitochondria”, first paragraph, Figure 6E). TEM assay was a concern of reviewer #2 and was not addressed in the response to reviewers. Figure 6C looks very strange (huge, probably lysed, areas around "potential" autophagosomes). I would honestly remove TEM experiments and replace them either with Western blots showing degradation of mitochondrial markers (HSP60, Tim50) or indeed use use mito-keima flux assay, as suggested by reviewer #2 (with tools already commercially available). That said I could also suggest to authors that for this paper they might omit description of results concerning mitophagy and address them in another project, in a future paper. This can be an opportunity to study the role in AURKA in mitophagy regulation in more details, more properly. This manuscript can stand on its own even without mitophagy.

Reviewer #3:

In the manuscript by Bertolin et al. the authors provide some evidence supporting a role for Aurora A (AURKA) kinase in control of mitochondrial function, including ATP production, membrane dynamics and mitophagy. Thus, this work could establish a novel role for AURKA in interphase cells, as opposed to well established mitotic function of this protein. Some data (e.g. mitochondrial localization of AURKA) are convincing, with appropriate controls, and well discussed. However, the majority of this work relies on correlative evidence, and mechanistic insights into how AURKA control mitochondria are missing. There are some inconsistences in data interpretation, and many conclusions (especially regarding mitophagy) are not supported by high quality data. Furthermore, some techniques used in this work should be supported by a more appropriate quantitative methods (especially mitophagy analyses).

1) While convincing, the FRET/FLIM data (Figure 2) should be supported by additional controls. Can the authors provide the evidence that other abundant OMM proteins do not show fluorescence lifetime alterations when co-overexpressed with AURKA?

2) The fact that overexpressed AURKA shows abundant cytosolic localization does not support the notion that AURKA is reexported from the mitochondria to the cytosol, as proposed (subsection “AURKA localises in the mitochondrial matrix via an N-terminal MTS and it undergoes a double proteolytic cleavage”, fifth paragraph). Many mitochondrial proteins when overexpressed localize to the cytosol, but rather through defects in protein import, than their export to the cytosol. The issue of AURKA export from the mitochondria, while potentially interesting, should be supported by stronger data. For example, can the authors use photactivable -GFP tagged AURKA, or some photobleaching assay to test this issue more directly?

3) It is surprising that some data on mitochondrial function in AURKA knockdown and overexpression were generated HEK293 or MCF7 cells, but mitochondrial fusion assays shown in Figure 4 were done using *Drosophila* cells. Can the authors justify why they do not move mammalian cell data (currently Figure 3—figure supplement 4) into the main figure?

4) The authors claim a direct role for AURKA in mitochondrial dynamics. However, these conclusions are based on mostly correlative data, and the possibility that other events affected in AURKA RNAi and AURKA-overexpressing cells contribute to this process, and the observed effect is not direct. Ideally, more data on directness of this mechanism would make this work much stronger. However, if this is not possible, then at least the authors should tone down their conclusions.

5) Generally, the mechanism by which low and high levels of AURKA show distinct effects on mitochondrial fusion and fission should be considered with more depth and supported by some additional evidence. Data provided in a current version of this work make it difficult to evaluate the underlying mechanisms.

6) The imaging data on mitochondrial morphology in four breast cancer cell lines is less convincing. For example, in case of MDA-MB-2311 cells one panel (DMSO-treated cells) show small cells with perinuclear mitochondria, the other one (MLN8237-treated cells) shows multinuclear large cell. Furthermore, inserts showing magnified mitochondria in each cell type would make data analysis/evaluation much easier. The authors should also show some additional data testing the possibility that the underlying mechanism is indeed conserved (perhaps Drp1 phosphorylation levels).

7) The mitophagy data is not very convincing. In addition to measuring mitochondrial mass/area, more direct mitophagy assays (e.g. mito-Keima) should be performed to verify the effects of AURKA expression on this process. Furthermore, the authors claimed that overexpression of AURKA directly regulates mitochondrial fusion (e.g. Figure 4). They also stated that there are changes in expression of some fusion proteins (relatively minor). However, Figure 6 shows ~50% decrease in mitochondrial mass. Why such dramatic differences in fusion protein are not observed in Figure 4. Is Opa1 somehow spared from degradation? This is very confusing and should be addressed. For example, total cell lysates should be analyzed for expression levels of many distinct mitochondrial proteins (and non-mitochondrial controls) and their changes in AURKA-depleted, AURKA-overexpressing and control cells quantified. Furthermore, why the authors do not analyze LCIII localization in mammalian cells, but instead use *Drosophila*?

8) LCIII is not cleaved upon activation of mitophagy (as stated here), changes in LCIII mobility are due to conjugation to the head group of the phosphatidylethanolamine. This should be corrected.

---

## [Author Response]

[Editors’ note: the author responses to the first round of peer review follow.]

As you will see in our point-by-point response, we have carefully addressed each of their concerns with new figures throughout the manuscript, detailed explanations or with a more appropriate discussion of our data. These new elements are now highlighted in the manuscript. We are confident that you and the reviewers will find in our revised manuscript convincing answers in response to the key points raised.

Reviewer #1:[…] 1) The study is overall compelling and well written, with helpful diagrams. One point that is confusing is the mechanism by which AURKA may function. This may go beyond the depth of this study, however, it makes the findings that both overexpression and loss of AURKA lead to longer mitochondria somewhat confusing. It would be useful (at least in the Discussion) to address how both loss and gain of function could lead to similar phenotypes. Does this mean that normally, AURKA is controlling mitochondrial quality by degrading non-functional mitochondria but when overexpressed it increases fusion of functional mitochondria. This point is interesting but confusing.

First, we would like to thank Dr. Rosenblatt for her positive and helpful comments on the content and on the style of our manuscript.

To answer the first comment:

The similarities

In the case of a loss of function of AURKA there is a lack of fission, and as a consequence the mitochondria network looks highly interconnected. In the case of a gain of function/overexpression there is an increase in fusion and in mitophagy that eliminates small damaged mitochondria. Again, the mitochondria network look highly interconnected.

The differences

In the case of AURKA loss of function, the mitochondrial mass and the ATP production remain unchanged compared to controls. In the case of gain of function, there is a loss of mitochondria mass (around 40%) due to mitophagy and an increase in ATP production.

To shade light on this apparent discrepancy concerning the morphology of the mitochondrial network, we provide additional results and we discuss them more in depth in the corresponding Discussion section. In agreement with a similar request from reviewer #3 (c.f. point 2 of the response to reviewer #3), we provide new evidence that cells where AURKA is inhibited with MLN8237 show no differences in mitochondrial respiration compared to controls (new Figure 3L, subsection 2 Over-expressed AURKA increases the abundance of mitochondrial complex IV and up-regulates ATP production”, second paragraph). This is in agreement with the similar effect of *AURKA* gene silencing on mitochondrial respiration (Figure 3H). However, MLN8237 induces mitochondrial elongation in four carcinoma cell lines routinely used for cancer cell biology (new Figure 5B, subsection “AURKA regulates mitochondrial fusion when over-expressed”, third paragraph), as treatment with an AURKA-specific siRNA does. Although it is true that the morphology of mitochondria in cells treated with MLN8237 is comparable to the one of cells overexpressing AURKA, in the case of overexpressed AURKA a positive selection of mitochondria (through mitophagy and fusion) enhances mitochondrial ATP production.Instead, at physiological levels, AURKA promotes mitochondrial fragmentation and this is revealed by an increased mitochondrial interconnectivity when AURKA is silenced or inhibited. Under these conditions, there is no positive mitochondrial selection and thus no increase in ATP production. This is in agreement with data from Kashatus et al. (2011), where a dominant-negative version of AURKA also promotes mitochondrial fusion. We hope that these new results will help clarifying the action of AURKA on mitochondrial morphology according to its abundance.

a) With respect to this point, what does this sentence mean? "These results suggests that AURKA maintains mitochondrial fission when expressed at physiological levels, and in its absence a compensatory fusion could be established as a consequence of lack of fission."

We thank the reviewer for this comment. We modified the sentence as follows: “These results indicate that AURKA maintains mitochondrial fission when expressed at physiological levels and that mitochondrial interconnectivity in the absence of AURKA is a consequence of a lack of fission. This results in the mere accumulation of elongated mitochondria without any increase in the energetic capabilities of the mitochondrial network”. We hope that this new statement improves the clarity of the conclusion.

2) Since there is much discussion about the role of increased AURKA function in cancer, they should provide more data regarding the prevalence of AURKA overexpression or mutations in different cancers. How common is it? Is it an indicator of poor prognosis or restricted to certain types of cancers? Do you see similar mitochondrial hallmarks in these cancers or do other mutations alter the phenotypes?

At present, increased copy number of the AURKA gene region is generally associated with an aggressive disease and poor patient survival. The AURKA gene region is located on chromosome 20, and its amplification includes the enhanced expression of additional genes (e.g. genes regulating cell cycle progression, and the most well-described AURKA interactor TPX2) (Belt et al., 2012; Sillars-Hardebol et al., 2011). In addition, the overexpression of AURKA has been linked with chromosomal instability (Baba et al., 2009). These events are common in different cancer types as in ovarian, pancreatic, lung and colon cancers and lead to bad prognosis. For instance, the increased copy number of AURKA is associated with the evolution of colorectal polyp into carcinoma (Carvahlo et al., Gut 2009). In breast cancer, the overexpression of AURKA is also linked to poor survival and it is associated with the overexpression of the human growth factor receptor 2 (HER2) and progesterone receptor (Nadler et al., 2008). Although epithelial cancers are non-glycolytic tumours and use the OXPHOS chain to produce ATP (Whitaker-Menezes et al., 2011), none of the above-mentioned studies took into account mitochondrial dysfunctions caused by or appearing in the presence overexpressed AURKA. Our study is thus pioneer in correlating for the first time this multifaceted kinase and mitochondrial physiology. This paragraph has been inserted in the Discussion, first paragraph.

We also provided additional data concerning four different breast cancer cell lines (other than MCF7), where AURKA is differentially abundant (New Figure 5, subsection “AURKA regulates mitochondrial fusion when over-expressed”, third paragraph). All these carcinoma cell lines show the recruitment of AURKA at mitochondria and the interconnectivity of the mitochondrial network is positively correlated with the abundance of AURKA.

We hope that these explanations together with the additional data provided will now be judged sufficient.

Reviewer #2:[…] In my opinion, the data described here is not really believable nor is its physiological significance clearly demonstrated. The entire story rests on there being a pool of AURKA that is imported into mitochondria. Even this conclusion seems dubious. AURKA isn't found in either Mitocarta or Mitominer, suggesting that dozens of experiments integrated to make these databases have failed to identify AURKA in this compartment. Although there could of course be proteins that are mitochondrial that have been missed as being mitochondrial, the absence of AURKA in combination with the absence of a clear mitochondrial targeting sequence in its N-terminus raises questions.

We agree with the reviewer that the MTS of AURKA is rather unconventional. There is no aminoacidic consensus or similarity when we compared it to the most extensively described MTS in the literature. It is also true that the software predicting the existence of a putative MTS in public databases (e.g. MitoCarta, Mitominer) mostly fail in recognising AURKA as a mitochondrial protein. Despite this, it is known that there is no “standard” MTS sequence or consensus (even fully characterised MTS show a quite high degree of plasticity), that MTS differ according to the final intramitochondrial localisation (Reviewed in Chacinska et al., 2009), and that the majority of this prediction software is set to recognise proteins targeted to the matrix with a higher affinity than the other classes of sub-mitochondrial proteins. For all these elements, AURKA might have failed being classified as a mitochondrial protein. However, we believe that we provide sufficient elements in our manuscript to support the conclusion that AURKA is indeed mitochondrial.

Remarkably, it has recently been published that a pool of the kinase localises at the kinetochore, although public databases failed in detecting AURKA as a kinetochore-bound protein (Courthéoux et al., J Cell Sci 2018). Therefore, it is also conceivable that a pool of AURKA is mitochondrial, without it being recorded so far.

We indeed acknowledged that the MTS of AURKA is not conventional. We were indeed very surprised by the fact that unlikely the majority of “conventional” MTS, this one allows the mitochondrial import of the protein even with a fluorophore at its N-terminus (Figure 2D and E, subsection “AURKA is enzymatically active at the mitochondria”, first paragraph). This strongly suggests that the conventional MTS located at the N-terminus are not the rule and that the amino acid sequence might not be the sole criterion for importing a protein into mitochondria. Other criteria such as the structure of the protein or its post-translational modifications could be involved. In this light, our molecular modelling approach provides evidence that changes in the three-dimensional conformation of the MTS (e.g. when comparing normal AURKA or the cancer-related variant F31I, Figure 8A and B, subsection “The cancer-related AURKA Phe31Ile is more imported into mitochondria and it accelerates mitophagy and ATP production”, first paragraph) impact the import of the kinase.

The fact that AURKA is imported into mitochondria and that it bears a “cryptic” and unconventional MTS has also been reported by a competitor team in a recent BioRXiv preprint (Grant et al.), while our manuscript was under review.

One must then wonder how the authors either see signals for AURKA overlapping with mitochondria or the presence of processed AURKA in extracts, with the processed form diminished upon depletion of the supposed mitochondrial protease PMPCB. One possibility for the processing seen in extracts is that cell lysis results in release of the protease from the mitochondria that has the ability to cleave AURKA, a possibility that wasn't tested. The import model is seemingly made even more unlikely in that they claim that a GFP-AURKA-Cherry fusion is also recognized and imported into mitochondria. It is hard to see, given how the translocon and mitochondrial target sequences work, that this construct would be successfully imported into mitochondria and cleaved by PMPCB.

We understand the reviewer’s concern. The reviewer is concerned by the fact that AURKA might be cleaved during the process of protein extraction that put together the kinase and mitochondria PMPCB protease. The import signature, intended as two bands on immunoblots which reflect a double cleavage inside mitochondria, could thus reflect a non-physiological protease activity of PMPCB, acting only in the extract. To convince the reviewer, (1) we are using standard procedures to extract mitochondria, thoroughly employed in the literature (2) we performed additional total protein lysis using Laemmli sample buffer. As the reviewer knows, the advantage of using Laemmli as a lysis buffer is that it induces a complete inhibition of all enzymatic activities while lysing the cells. Under these conditions, the AURKA_38_ band is still present in cells transfected with a control siRNA. This indicates that AURKA_38_ band is not due to the mitochondrial fractionation procedure (see Author response image 1, red arrowhead), but it is present in the cell.

**Author response image 1. respfig1:** Total lysates from HEK293 cells obtained by immediate lysis in Laemmli sample buffer. Cells were blotted for endogenous AURKA, TUBA1A and PMPCB. AURKA_46_ and AURKA_38_ are indicated with a blue and red arrowhead, respectively.

The authors perform essentially every mitochondrial assay available to them to provide evidence of effects on mitochondrial function, either by depletion of AURKA or overexpression (at leaves that appear to be ~100-fold or more higher than endogenous). Many of these assays end up being descriptive, and they paint a bewilderingly complex interplay between AURKA and mitochondria, which makes it hard to put any specific finding into a logical biological process.

The reviewer also complains about the number of assays done to investigate the role of AURKA at mitochondria, which is too elevated in his opinion. We were very surprised by this comment and we believe that this is not the case, as only by screening multiple and diverse parameters of mitochondrial functionality we could see the action of physiological AURKA in maintenance of mitochondrial morphology, and the one of overexpressed AURKA in ATP production and autophagy. Given that reviewer #3 asks for more data in support of the role of overexpressed AURKA in ATP production (c.f. point 2 of the response to reviewer #3), it is clear that the number of assays claimed to be excessive were actually not sufficient.

As an example, the authors employ immunoprecipitation of overexpressed AURKA-GFP followed by mass spectrometry and identify 2600 proteins (approximately 25% of the expressed proteome in these cells), including 449 mitochondrial proteins. They apparently have no negative controls and try to make the conclusion that because certain mitochondrial proteins are detected, that they are likely real and direct. It is well known that the vast majority of proteins present on anti-GFP resin are non-specific binders, but they haven't tried to control for this. So ultimately, they do not rigorously demonstrate that the proposed interactions with proteins that are both in the inner membrane and function at the outer membrane during fission are specific or not. Thus, it becomes very cloudy in terms of how AURKA is working.

We thank the reviewer for this comment. The mitochondrial proteome of AURKA obtained by MS/MS data that we presented in the first version of the paper corresponded to the raw data without any cut-off procedure. We now inserted stringent filter to present our data. First, we eliminated all the proteins corresponding to only one or two peptides in the MS/MS data. Then, we filtered our data by excluding common contaminants non-specifically binding to anti-GFP beads and present in the CRAPome database. All the experimental details were updated in the new version of the manuscript, in the Mass spectrometry section of the Materials and methods. We now have a list of 641 putative AURKA-interacting proteins in all cellular compartments, as reported in the new Supplementary file 2. Again, we noticed that mitochondrial proteins are overrepresented (new Supplementary file 2 and Figure 5—figure supplement 7; Discussion, third paragraph).

Is its main function(s) within the matrix or in the cytoplasm in the context of fission with DNM1L/MFF?

Given the multiple functions of overexpressed AURKA in the mitochondrion, and asks which one is the predominant. A wave of reports in the last two decades revealed that changes in mitochondrial dynamics and mitophagy are tightly intertwined. Changes in mitophagy cannot occur in the absence of dramatic alterations in the morphology of the mitochondrial network and in its energetic capabilities. Given that AURKA can act on three levels – mitochondrial dynamics, mitophagy and ATP production -, it is hard to discriminate which change comes first. Most likely, there is a continuum of undetectable micro-changes happening simultaneously; with the tools available nowadays, these changes become visible only when they are already broadly activated. In this light, the development of new tools to detect more subtle modifications in mitochondrial physiology will represent a significant breakthrough in the field in the next years. In addition, the functional characterization of the AURKA interactome is certainly a goal in understanding the role of mitochondrial AURKA more precisely. However, we feel this is beyond the scope of the present paper.

For the vast majority of experiments, the authors use either a single siRNA or a single shRNA. Thus, there are no experiments to demonstrate the absence of off target effects for any individual RNAi reagent. Given that many of the effects are small, it is conceivable that some of the conclusions drawn are based on off-target effects.

The reviewer is also concerned by the validity of our siRNA/shRNA approach. We agree that we did not perform any rescue of these siRNA/shRNA experiments. We would like to underline that our results are supported by the data obtained in the *Drosophila* model (Figure 4 and Figure 3—figure supplement 4;subsection “AURKA regulates mitochondrial morphology”, last paragraph). In this organism we employ mutants inducing a full knock-out of the protein to analyse the mitochondrial network connectivity and mitophagy. We believe that this parallel approach is much more relevant than siRNA/rescue. In addition, we also employ the pharmacological inhibitor of AURKA MLN8237 to support our conclusions on the role of AURKA in mitochondrial morphology (new Figure 5B, subsection “AURKA regulates mitochondrial fusion when over-expressed”, third paragraph).

Additionally, in Figure 3 the authors conclude that high levels of AURKA lead to increases in Complex IV, but no substantial changes in other electron transport chain complexes. The weird thing here is that siRNA against AURKA also lead to slight increases in Complex IV levels, which doesn't make sense. One would think there would be opposite activities. The physiological significance of most of the measurements in this figure isn't clear.

The reviewer asks why only Complex IV of the mitochondrial respiratory chain is increased in conditions of AURKA-overexpression. This scenario is actually quite common in the field of mitochondrial disorders, where slight deficiencies in single complexes (often due to mtDNA mutations) can act alone in decreasing the functionality of the entire mitochondrial respiratory chain. This gives rise to complex, early-onset myopathies and neurological disorders (reviewed in Schapira, Lancet 2012). Although AURKA does not appear to be an integral protein of the mitochondrial respiratory chain, we believe that its interaction with several subunits of this chain could participate in producing excessive quantities of ATP.Of note, the siRNA against AURKA does not increase Complex IV levels as shown in Figure 3A and in the corresponding quantification (subsection “Over-expressed AURKA increases the abundance of mitochondrial complex IV and up-regulates ATP production”, first paragraph).

Many of the conclusions just don't make sense and could result from overexpression effects. For example, they state "Last, no LAMP1 staining was observed in cells over-expressing AURKA ΔNter (Figure 6C), strongly suggesting that the mitochondrial pool of AURKA is responsible for mitophagy activation." It makes no sense that overexpression would completely remove LAMP1 staining in cells. Does this suggest that there are no lysosomes, for example? The authors do not look any further than this but make the conclusion that AURKA regulates autophagy – a complete non-sequiter.

The reviewer is concerned by the assays we used to determine the impact of AURKA in the elimination of dysfunctional organelles by mitophagy. The first question raised is on the lack of LAMP1-labeled lysosomes in AURKA ΔNter-transfected cells and on whether this means that there are no lysosomes in these cells. To answer this concern, we analysed the fluorescence intensity of LysoTracker Red by flow cytometry in cells transfected with an empty vector or with AURKA ΔNter (newFigure 5—figure supplement 1B; subsection “fluorescence measured by flow cytometry. n=3 independent experiments with at least 30,000 cells per”, second paragraph). No difference was detected in the two conditions, showing that lysosomes are equally present and unaffected in their abundance. The absence of LAMP1 in cells transfected with AURKA ΔNter might be the readout of an impaired or slowed targeting of lysosomes to mitochondria, or being linked to defective lysosomal composition. In both cases, this could result in defective mitochondrial degradation by mitophagy. The lysosomal composition and the correct targeting of lysosomes have already been described to play an important role in paradigms of cytotoxicity mediated by Natural Killer cells (Kzrewski et al., Blood 2013). These two aspects and their dependency on AURKA have not been evaluated in our study. Although very interesting, we believe this falls out of the scopes of our paper.

Further evidence that AURKA does not only impact LAMP1 abundance but global mitochondrial mass is also presented in the paper, underlining that LAMP1 is not a preferential/unique target of AURKA. It should rather be considered as an event falling in a pathway which perturbs mitochondrial functionality.

Finally, the authors try to link AURKA to mitophagy. They claim they use accepted assays but this really isn't the case. The TEM assay is very difficult to quantify, and overall they are looking at very small numbers of cells and trying to make conclusions. It isn't clear why they didn't use either the accepted mito_keima flux assay or alternatively use anti-DNA immunofluorescence, which allows quantification of the amount of mitochondria based on the mitochondrial DNA abundance. Both of these assays are highly used in the field.

We thought our assays are standard in the field, as reviewed in Klionsky et al., 2016 (electron microscopy, LC3 cleavage assessed by biochemical approaches, fluorescence microscopy to localise LC3 and LAMP1 to mitochondria, to calculate the fluorescence intensity of an LC3-timer construct, and to quantify mitochondrial mass).

To follow the suggestion of the reviewer, we added an additional assay relying on the quantification of the fluorescence intensity of MitoTracker Green by flow cytometry (new Figure 6G and 8I; subsection 2 Over-expressed AURKA triggers mitophagy of fragmented mitochondria”, first paragraph and subsection “The cancer-related AURKA Phe31Ile is more imported into mitochondria and it accelerates mitophagy and ATP production”, third paragraph). Given that this probe is insensitive to the polarization status of mitochondria, it gives a direct readout of the mitochondrial mass. MitoTracker Green confirmed the existence of mitochondrial loss when AURKA is overexpressed. As this probe is very common in the field, we hope that the reviewer will be convinced by the fact that AURKA triggers mitophagy when overexpressed. We also added further experimental evidence showing that cells treated with the autophagy inhibitors 3-methyladenine and Bafilomycin A1 block mitophagy in cells overexpressing AURKA (new Figure 7C; subsection “Over-expressed AURKA triggers mitophagy of fragmented mitochondria”, second paragraph). Overall, we hope that the reviewer will be reassured that AURKA induces mitophagy when overexpressed.

Based on these and many other issues with the paper, I cannot support publication in eLife. I think a much more rigorous analysis would need to be done. Moreover, if AURKA is functioning as a kinase in the mitochondria, what are its substrates and how does phosphorylation by AURKA alter their functions.

Although we obtained a proteomic analysis of the AURKA interactome, we agree with the reviewer that key mitochondrial substrates of AURKA need to be functionally validated. However, we feel that this will be the focus of future work, and it falls out of the scopes of the present manuscript.

We hope that our answers and additional experiments provided will reconcile the reviewer with our work and convince him/her of its validity.

Reviewer #3:[…] In general, the data showing mitochondrial localization of Aurora A are solid and highly interesting and it is also clear that AurkA overexpression induces significant morphological and functional changes in mitochondria. Major remaining question is whether these changes are a consequence of the a) AurkA-NML1 interaction, b) the established RALA-DNML1-dependent pathway that controls mitochondrial fission/fusion or c) a fusion/fission-independent pathway as suggested by the Phe31Ile mutant. Some of these questions are acknowledged in the Discussion but it seems that the information in the manuscript remains a bit preliminary until some of these questions are addressed more in detail.

We thank reviewer #3 for their thoughtful comments on our manuscript. We feel that their suggestions greatly improved the overall quality of our study in this present version.

1) The localization of AurA seems to be variable in the different images shown. For instance is only cytoplasmic in Figure 1 (or Figure 2D) with certain spotted (mitochondria) pattern, but is both cytoplasmic and nuclear in Figure 2B, C with no evidences of spotted (mitochondrial pattern). One possibility is that cell in Figure 1 is G1 whereas it might be S/G2 in Figure 2. The authors should clarify the pattern of Aurora A localization (and co-localization with mitochondria) in the different cell cycle stages.

We thank the reviewer for this comment. The differential localisation is most likely due to the expression levels. In Figure 1 and 2D, AURKA is expressed under its endogenous promoter, which yields physiological levels of the kinase in the cell. Conversely, in Figure 2B/C the kinase is strongly overexpressed and a stronger labelling in the cytosol is indeed present. That said, we also explored whether endogenous AURKA shuttles to mitochondria in a specific cell cycle phase. Of note, this aspect was also raised by reviewer #1 (c.f. point 3 of the response to the reviewer #1). These results were included in the new version of the manuscript (new Figure 1—figure supplement 1A; subsection “AURKA localises in the mitochondrial matrix via an N-terminal MTS and it undergoes a double proteolytic cleavage”, first paragraph), and show that AURKA shuttles to mitochondria in any cell cycle phase and regardless to its abundance. We hope that the reviewer will be satisfied with this new finding.

This cytosolic staining shows AURKA before its import into mitochondria, but also the mitochondrial form exported from the organelles. These two forms, which can be distinguished on Western blotting as shown in Figure 2A, are indistinguishable on immunofluorescent micrographs.

2) It is somehow surprising that OCR does not change after Aurora A knockdown. I wonder whether the knockdown levels are not sufficient. The authors should also use AurkA inhibitors to test mitochondrial activity (Seahorse).

We thank the reviewer for this comment, which allowed us to go further in our understanding of the mitochondrial changes induced by AURKA. Following their suggestion, we explored OCR in cells treated with an acute (250 nm, 10 min) or prolonged (100 nM, 3 h) dose of MLN8237, and we did not observe OCR variations in any of the conditions analysed (new Figure 3L; subsection “Over-expressed AURKA increases the abundance of mitochondrial complex IV and up-regulates ATP production”, second paragraph). This is in agreement with the similar effect of *AURKA* gene silencing on mitochondrial respiration (Figure 3H; see the aforementioned paragraph). Although MLN8237 induces mitochondrial elongation in four carcinoma cell lines routinely used for cancer cell biology (new Figure 5B; subsection “AURKA regulates mitochondrial fusion when over-expressed”, third paragraph), an elongation morphologically comparable is observed in cells overexpressing AURKA. In the case of overexpressed AURKA a positive selection of mitochondria (through mitophagy and fusion) enhances mitochondrial ATP production. Instead, at physiological levels, AURKA promotes mitochondrial fragmentation as revealed by an increased mitochondrial interconnectivity when AURKA is silenced or inhibited. Under these conditions, there is no positive mitochondrial selection and thus no increase in ATP production is required. This is now more clearly stated in the manuscript, subsection “AURKA regulates mitochondrial morphology”, first paragraph. We hope that the reviewer will be satisfied with these new findings and will agree on the fact that they further help the understanding of the roles of AURKA into mitochondria.

3) On a related issue, how did the authors manage to prevent secondary consequences of cell cycle arrest/defects in these assays? Does changes in mitochondria induced by AurkA occur in parallel to changes in cell cycle phases (entry into G2 etc.)?

Following up on the new results described in the first point, AURKA localises to mitochondria in all cell cycle phases (Figure 1—figure supplement 1A; subsection “AURKA localises in the mitochondrial matrix via an N-terminal MTS and it undergoes a double 85 proteolytic cleavage”, first paragraph). This supports the fact that its activity into mitochondria is constant throughout the cell cycle, and not only at cell cycle checkpoints.

4) It is not clear whether the authors are proposing a direct activity of Aurora A on DNML1-Ser637? Or changes in Ser 637 are RALA-dependent following the already established pathway? A direct interaction between AurkA and DNML1 is suggested by the proteomics data but the authors do not evaluate this functionally. One would assume that is a RALA-dependent process but his has not been tested and RALA apparently was not present in these complexes. This mechanism should be analyzed in detail and properly clarified. Perhaps rescue assays with RalA, DNML1 would help.

Again, we thank the reviewer for this comment, which helped us in showing that AURKA acts on two different pathways: (1) AURKA phosphorylates RALA to trigger its mitochondrial localisation (RALA-dependent pathway, described in Kashatus et al., 2011); (2) AURKA enters mitochondria in a RALA-independent pathway. First, RALA is not required for the import of AURKA inside mitochondria (new Figure 1—figure supplement 3E; subsection “AURKA localises in the mitochondrial matrix via an N-terminal MTS and it undergoes a double proteolytic cleavage”, fourth paragraph), according to our new data. Second, a version of AURKA (AURKA ΔNter) which can phosphorylate RALA on Ser194 (Figure 3—figure supplement 6D; subsection “AURKA regulates mitochondrial fission in physiological conditions”, first paragraph), but is not importable into mitochondria, was not capable of reverting the mitochondrial connectivity phenotype observed when endogenous AURKA is depleted by siRNA (new Figure 3—figure supplement 6C; see the aforementioned paragraph). This demonstrated that it is the presence and the function of AURKA into mitochondria that are required to maintain mitochondrial morphology. We included these data in the new version of the manuscript, and we hope that these findings will satisfy the reviewer. Additional FRET data not included in the manuscript showed that the physical interaction between AURKA and DNM1L does not depend on the presence of RALA, as it is retrieved in cells treated with a control- or with a RALA-specific siRNA. We attach these results in Author response image 2.

**Author response image 2. respfig2:** Fluorescence lifetime quantification of MCF7 cells transfected with AURKA-GFP and an empty vector (-) or DNM1L-mCherry, and silenced or not for RALA. *** P<0.001; NS: not significant.

We found that the fission-prone function of AURKA expressed at physiological levels (RALA-independent pathway) goes through the phosphorylation of DNM1L on Ser637 (Figure 3—figure supplement 6B; subsection “AURKA regulates mitochondrial fission in physiological conditions”, first paragraph), as indicated in the first version of our manuscript. Conversely, the RALA-dependent pathways relies on the phosphorylation of DNM1L on Ser616 (Kashatus et al., 2011). Therefore, data provided on the phosphorylation status of DNM1L further support the existence of two distinct pathways. We hope that the existence of a RALA-dependent and -independent pathway for the functions of AURKA at mitochondria will satisfy the reviewer.

5) Data with the Phe31Ile mutant are quite interesting and this mutant has a significant effect in mitophagy and mitochondrial activity. However, given the lack of specific mechanism as discussed in the previous point, it is not completely clear why this mutant induces these effects. The authors indicate that the mutations favors entry into the mitochondria; however, no changes in interconnectivity are shown. As mitophagy was previously proposed to be triggered or favored by these changes in connectivity, the actual reason for the phenotype induced by the AurkA mutant is not really clear. In addition, these data actually argue against the DNML1-fusion mechanism proposed in the earlier figures in the manuscript.

As explained in the previous point, we now show that the role of AURKA into mitochondrial falls out of its interaction with RALA. Therefore, the reason why AURKA Phe31Ile has an even more drastic effect on mitophagy and energy production than normal AURKA must be searched into the intrinsic characteristics of the mutant. By molecular modelling, we show that the MTS adopts a different conformation when Phe31 is mutated into an Ala. This change makes AURKA more suitable to enter mitochondria.

As the reviewer observes, it is true that there are no differences in terms of mitochondrial connectivity between normal and mutated AURKA. The new findings inserted in the paper (new Figure 3L and 5B; subsection “Over-expressed AURKA increases the abundance of mitochondrial complex IV and up-regulates ATP production”, second paragraph and subsection “AURKA regulates mitochondrial fusion when over-expressed”, third paragraph) now underline that the connectivity is a poor readout to estimate the energetic capabilities of the mitochondrial network in the paradigm of AURKA. Indeed, AURKA Phe31Ile enhances mitophagy and energy production more importantly than normal AURKA, although the mitochondrial connectivity degree is comparable. Future studies will certainly elucidate the complex scenario of the AURKA F31I interactome, but this is not the focus of the present manuscript.

[Editors' note: the author responses to the re-review follow.]

On the whole, there was consensus that the paper was much improved. However, some remaining issues were identified that need to be addressed before acceptance, as outlined below:All three reviewers found the data on mitophagy not particularly convincing. There was also concern that the cancer data was not as well-developed as it might be with respect to rescuing the mutation and imaging. Therefore, we suggest that you also omit this part of the story unless you feel that it can be better supported. These are our only requests required for acceptance. I include the comments of the reviewers below only for your own reference and feedback.

First, we would like to thank the three reviewers and the reviewing editor for their constructive comments, and for helping us to see our manuscript from another perspective.

There was only one request by the editors for the acceptance of the paper: removing the parts concerning mitophagy and the cancer-related mutation of AURKA Phe31Ile.

For this version of the manuscript, we followed the editorial requests and removed the sections concerning mitophagy and the cancer-related AURKA polymorphism Phe31Ile.